# Regulating Retinoic Acid Availability during Development and Regeneration: The Role of the CYP26 Enzymes

**DOI:** 10.3390/jdb8010006

**Published:** 2020-03-05

**Authors:** Catherine Roberts

**Affiliations:** 1Developmental Biology of Birth Defects, UCL-GOS Institute of Child Health, 30 Guilford St, London WC1N 1EH, UK; catherine.roberts@ucl.ac.uk; 2Institute of Medical and Biomedical Education St George’s, University of London, Cranmer Terrace, Tooting, London SW17 0RE, UK

**Keywords:** CYP26, retinoic acid, embryonic development

## Abstract

This review focuses on the role of the Cytochrome p450 subfamily 26 (CYP26) retinoic acid (RA) degrading enzymes during development and regeneration. Cyp26 enzymes, along with retinoic acid synthesising enzymes, are absolutely required for RA homeostasis in these processes by regulating availability of RA for receptor binding and signalling. Cyp26 enzymes are necessary to generate RA gradients and to protect specific tissues from RA signalling. Disruption of RA homeostasis leads to a wide variety of embryonic defects affecting many tissues. Here, the function of CYP26 enzymes is discussed in the context of the RA signalling pathway, enzymatic structure and biochemistry, human genetic disease, and function in development and regeneration as elucidated from animal model studies.

## 1. Introduction

Retinoic acid (RA) signalling is required in postnatal and adult life where it has various roles which include: neurogenesis, synaptic plasticity, learning and memory (reviewed [1,2]) and circadian rhythms and seasonality [3]; retinal gap-junction neuromodulation and rod and cone cell function, (reviewed [4]); immune response for T-cell differentiation, immunoglobulin production, homing of innate immune cells to the gut and production of pro-inflammatory cytokines, (reviewed [5]); roles in lipid/fatty acid formation/breakdown, adipocyte differentiation and remodelling and energy metabolism (reviewed [6]) and postnatal skeletal growth and homeostasis (reviewed [7,8]).

RA homeostasis to regulate signalling by controlling available RA levels is absolutely required in the developing embryo. Both reduced and excessive RA levels are injurious to embryonic development, producing similar embryonic defects, usually leading to embryonic lethality [9], (reviewed [10,11]). Availability of RA is regulated by a balance between synthesizing enzymes (retinaldehyde dehydrogenases-RALDHs) and metabolizing enzymes (Cytochrome p450 family 26-CYP26s). Whilst substantial progress has been made regarding the function of RALDHs during development, much less is known about the CYP26s. This review gives a brief overview of the RA pathway and function in development, already well-described in numerous reviews, for example [10,11,12,13,14,15,16] and then focusses upon the role of the RA-degrading CYP26 enzymes during embryonic development and in regenerative processes.

## 2. Retinoic Acid Synthesis and Receptor Signalling

The retinoic acid pathway is characterised by a number of cytoplasmic oxidative enzymatic reactions, some of which are required to synthesise RA from dietary precursors including Vitamin A and others which are required to metabolise RA to less biologically active forms. RA can act cell autonomously (autocrine) entering the nucleus to bind to hetero-dimeric receptors at target gene promoters altering transcriptional activity of the promoter. It also acts non-cell autonomously, functioning as a classic morphogen, diffusing across cell membranes establishing retinoic acid gradients within and across tissues (paracrine).

RA is a lipophilic molecule, manufactured during embryogenesis from maternal retinol (vitamin A) in placental species and carotenoids in the yolk of oviparous species [17]. This process (summarised in Figure 1) begins with circulating retinol being taken up by embryonic retinol binding protein 4 (RBP4). This complex then binds to STRA6 (stimulated in retinoic acid 6) and is transported across the plasma membrane in the cell. *Stra6* is expressed in many tissues in the developing embryo, including the epithelia of the pharyngeal arches and facial mesenchyme. Human mutations in *Stra6* produce a complex phenotype including anophthalmia, lung hypoplasia, mental retardation and craniofacial and heart defects reminiscent of 22q11 Deletion Syndrome [18]. 

Retinol then must be converted by oxidation into the intermediate form of retinaldehyde which is accomplished by two enzyme families, the cytosolic alcohol dehydrogenases (ADHs) and microsomal retinol dehydrogenases (RDHs).

*Adh1*, *Adh3* (also known as *Adh5*) and *Adh4* (also known as *Adh7*) null mutations in mice are viable post-natally on a standard diet. When challenged by administration of retinol, levels of RA in the test tissue of the kidney increased in wild type mice. Lesser increases were seen for *Adh1^−/−^* and *Adh4^−/−^* animals. Further examination of *Adh3^−/−^* mice revealed slightly smaller litter sizes and reduced growth compared to wild type. On a vitamin A deficient (VAD) diet 100% of *Adh3^−/−^* eventually died, with 80% lethality between P0 and P3. For *Adh1^−/−^* mice 40% lethality by P40 was observed whereas *Adh4^−/−^* mice all died by P15. Growth deficiency was observed for all *Adh* mutations on a VAD diet but was more severe in *Adh3* and *Adh4* null mutant mice. Interestingly, double null mutants for *Adh1/4* exhibited a slightly milder phenotype than *Adh4^−/−^* animals alone, surviving slightly longer into the postnatal period with 100% penetrant lethality by P24. The VAD diet also affected embryonic viability. Live-born mice were seen in only 15% of *Adh1^−/−^* pups compared to 49% for wild-type and the reabsorption rate at e12.5 was 69% compared to 30% for wild type. When these mice were fed on retinol supplemented diets, *Adh3^−/−^* and *Adh 4^−/−^* displayed high levels (95%) of postnatal survival, whereas for *Adh1^−/−^* mice, only 36% survived to adulthood, with 64% lethality between P0 and P3 [19,20,21,22,23,24]. Therefore, the ADHs may be variously, and possibly redundantly, involved in RA synthesis from retinol and play protective roles against the effects of excess or reduced retinol.

Work in the chick embryo also suggests that *Cyp1b1* a p450 cytochrome enzyme may be able to convert retinol to retinaldehyde and retinoic acid during neural development [25]. Whilst mouse null mutations appear normal during embryogenesis [26], human mutations are associated with congenital glaucoma, Peters anomaly [27,28] and Axel–Riegers Syndrome [29,30].

Various RDH genes coding for retinol dehydrogenase enzymes have also been implicated in the oxidative conversion of retinol to retinaldehyde. *Rdh5* and *Rdh8* mutations are associated with mild night blindness in humans and individual knockout mice for these genes also display delayed dark adaption phenotypes [31]. *Rdh8^−/−^; Abca4^−/−^* double mutants also exhibit progressive retinal degeneration from 4–6 weeks of age [32]. *Rdh12* function in photoreceptors is essential for the visual cycle and mutations have been linked to Leber congenital amaurosis and autosomal dominant retinitis pigmentosa [33,34] and childhood onset severe retinal dystrophy. *Rdh12^−/−^* mice are prone to light-induced photoreceptor apoptosis. *Rdh8^−/−^; Rdh12^−/−^* double mutants have a slowly progressing rod cone dystrophy phenotype. Thus, these enzymes appear to be important in the clearance of trans-retinal, which otherwise produces *N*-retinylidene-*N*-retinylethanolamine (A2E), a toxic substance known to contribute to retinal degeneration and photoreceptor cell death [31]. *Rdh13* is also important in retinal biology, as *Rdh13^−/−^* mice suffer from acute light-induced retinopathy via the mitochondrial apoptosis pathway [35]. It is also implicated in liver injury, apoptosis and fibrosis as mutant mice displayed fewer injurious responses in carbon-tetrachloride-induced liver injury compared to wild type [36,37].

*Rdh1* mutant mice on a VAD diet show reduced *Cyp26a1* levels combined with increased retinoid stores, suggesting compensation for the lack of retinaldehyde and therefore RA in these mice. They also display increased size and adiposity. Finally, double null mice of epidermal short-chain retinol dehydrogenases *Sdr16c5* and *Sdr16c6* have accelerated hair growth and enlarged meibomian glands, upregulated hair-follicle stem cell genes and 80% reduced RDH activity, consistent with these genes regulating RDH function in the skin [38]. 

RDH10 and DHRS3 have been shown to regulate this retinol/retinaldehyde conversion in a reversible fashion. *Rdh10* is expressed in specific and dynamic fashion in early development, including lateral plate, paraxial and cardiac mesoderm. Mutations of *Rdh10* produce phenotypes reminiscent of various *Raldh2* loss-of function mutations. Full loss-of-function alleles give rise to embryonic lethality at ~E10.5 with shortened anteroposterior axes, defects in embryo turning, dilated and un-looped hearts, small somites, and forelimb bud agenesis, similar to *Raldh2^−/−^.* Point mutations of *Rdh10* gave rise to an RA-deficiency-like phenotype lethal at E13.0 which included pharyngeal arch, vascular and cardiac malformations such as common arterial trunk (CAT), ventricular misalignment and poor trabeculation and atrial septal defects, akin to *Raldh2^−/−^* plus maternal RA-supplementation phenotypes. These defects can be partially rescued by the administration of maternal RA, although cardiac defects remain recalcitrant to this treatment. Maternal retinaldehyde supplementation, however, rescues all defects to a greater degree and allows the production of viable and fertile *Rdh10^−/−^* adult mice. RDH10 is therefore thought to be the main enzyme involved in embryonic retinol to retinaldehyde conversion [16,39,40,41,42,43]. DHRS3, a short-chain dehydrogenase/reductase enzyme, reverses this reaction converting retinaldehyde back to retinol. Deletion of *Dhrs3* leads to late embryonic lethality (E15.5/18.5) with abnormal axial, craniofacial (palatal) and cardiac (VSD, ASD, DORV) as a result of increased at RA activity development. A similar result is seen in zebrafish mutants and Xenopus morphants. Both *Rdh10* and *Dhsr3* are responsive to levels of RA, and each requires the other for full enzymatic activity ([15,44,45,46,47], reviewed [16]). Thus, the RDH10 and DHRS3 enzymes controlling the bi-directional retinol–retinaldehyde conversion act to regulate against retinoid deficiency and excess retinoid levels respectively.

The final step of the RA synthesis pathway requires the irreversible oxidation of retinaldehyde into *all-trans* retinoic acid ligand by the retinaldehyde dehydrogenase (RALDH) enzymes [48,49,50] (Figure 1). Three of these enzymes are active during development. All three genes are expressed around the developing eye. *Raldh3* is also expressed in the olfactory placodes, whereas *Raldh2* has a wider expression pattern encompassing domains in the head, somatic and splanchnic mesoderm and the limb [51,52,53]. Null mutations for *Raldh1* and *Raldh3* suggest that these two family members are only necessary for embryonic RA production in a minor fashion as mutant phenotypes affect only the development of the eye, ear, forebrain and frontonasal process [54,55,56,57] (Figure 2).

In contrast, *Raldh2*, expressed in caudal mesodermal tissues (Figure 2), provides RA to the majority of embryonic tissues. *Raldh2^−/−^* embryos which are deficient in RA, die at mid-gestation at E10.5 with a very severe phenotype encompassing failure of axial rotation, a shortened anteroposterior axis and frontonasal process, abnormal somitogenesis, small otocysts, lack of limb buds and a single medial un-looped dilated heart cavity with impaired atrial and sinus venosus formation, posteriorly expanded SHF markers and impaired ventricular cardiomyocyte differentiation. A similar set of malformations has been observed in the zebrafish *raldh2* mutant *neckless* and recapitulated in morpholino knockdowns. Maternal/exogenous RA administration is able to rescue much of this phenotype such that embryos survive until E13.5–14.5 [58,59,60,61,62,63,64,65,66,67]. ‘RA-rescued’ *Raldh2* null mouse embryos and hypomorphic null allele of *Raldh2* display similar phenotypes including with early abnormalities of PAA1-3 and disorganized migration of the neural crest and later characteristic heart and thymus malformations similar to those described for 22q11DS and *Tbx1* mutations [64,68,69].

Local control of available retinoic acid via expression of RALDH (synthesis) as described above and CYP26 (degradation) enzymes (addressed in detail below), is an important element in the regulation of RA distribution across the embryo and within specific tissues/cell types, particularly given that several retinoic acid receptors, (RARα RXRα and RXRβ) have broad/ubiquitous expression patterns and can compensate for each other (reviewed by [70,71,72]).

Once RA has been generated in the cell by the synthesizing pathway, it binds cellular retinoic acid binding protein 2 (CRABP2) and is transported into the nucleus and delivered to the retinoic acid receptors (RARs) [11,73,74] to facilitate autocrine signalling within the cell. RA can also signal in a paracrine fashion to neighbouring cells by diffusion, a function important for developmental patterning gradients (Figure 1 and Figure 2).

The three conserved RARs (α, β and γ) are members of the nuclear receptor superfamily and bind RA in partnership with one of their three retinoid X receptor (RXRα, β and γ) heterodimer binding partners. Both RAR and RXR subtypes contain a number of different isoforms. The activity of RAR-RXR heterodimers is probably mediated via binding of all-trans-RA to the RAR partner [75,76]. Several retinoic acid receptors, (RARα, RXRα and RXRβ), have broad/ubiquitous expression patterns in general, with the others (RARβ, RARγ and RXRγ) showing more complex, tissue-specific expression. Different heterodimer combinations can transduce the RA signal in many tissues and there is a high degree of functional redundancy. Usually at least two receptors must be deleted in concert to ascertain any developmental defects. For example, compound receptor mutations which produce abnormalities of pharyngeal and outflow tract similar to 22q11DS include various compound mutants of *RAR*α*/*β, *RAR*α*/*γ and *RAR*β*/*γ, compound mutants of RXRα with any of the RARs and *RXR*α mutants alone [71,72,77,78,79,80,81,82].

In the absence of ligand, the RAR/RXR heterodimers bind to specific motifs within the promoters of target genes, known as retinoic acid response elements (RAREs). These are repeats of consensus sequences 5′-(A/G)G(G/T)TCA-3′ or 5′-(A/G)G(G/T)(G/C)A-3′ spaced at 1, 2 or 5 bp intervals. Unliganded receptor binding allows recruitment of co-repressor complexes, e.g., Polycomb group complexes such as NCor/SMRT which make the DNA unavailable for transcription. When RA binds to the RAR, a conformational change is initiated in the RAR-ligand binding domain, which results in the co-repressor complexes being released. Transactivating complexes such as SWI/SNF, NF1, pCIP/p300 and PolII are recruited instead, leading to the induction of chromatin remodelling and the activation of the transcriptional machinery. Alternatively, ligand-dependent repression can mediated by binding of proteins such as NRIP1, PRAME and TRIM24 (reviewed by [10,11,83] (Figure 1).

Maternal diet-derived retinol/in blood/yolk sac/yolk of embryo bound to retinol binding protein 4 (RBP4) enters cellular cytoplasm via binding to membrane-bound RBP/RA complex receptor STRA6 (Stimulated by Retinoic Acid 6). Retinol is then bound to cellular retinol binding protein (CRBP) and reversibly oxidised to the intermediate form retinaldehyde (also known as retinal) by alcohol dehydrogenase and retinol dehydrogenase (particularly RDH10) enzymes. The reverse reaction, retinaldehyde to retinol, is catalysed by the dehydrogenase/reductase 3 (DHRS3) enzyme. Retinaldehyde can also be generated from β-carotene by β-carotene 15,15′-monooxygenase (BCO). Retinaldehyde is then irreversibly converted to retinoic acid (RA) by retinaldehyde dehydrogenase (RALDH) enzymes, particularly RALDH2. RA can then undergo three different processes: (1.) RA bound to cellular retinoic acid binding proteins which shuttle RA to the nucleus. Hetero-dimerised RAR-RXR complexes (retinoic acid receptor-retinoid-X-receptor) are bound to conserved retinoic acid responsive elements (RARE) within the promotors of target genes. Most frequently, in the absence of RA co-repressor complexes (e.g., NCoR/SMRT) are bound to the RAR-RXRs, preventing transcription. Upon RA binding, the receptors undergo a conformational change, releasing co-repressor complexes and recruiting co-activator proteins (e.g., SWI/SNF, pCIP/p300, PolII) as replacements, thus triggering transcription activation of target genes. (2.) RA produced in one cells can also signal in a paracrine fashion to neighbouring cells, mediating non-cell autonomous effects. (3.) If Cytochrome P450 subfamily 26 (CYP26) enzymes are present in the cell, RA is hydroxylised in the cytoplasm to more polar metabolites with less biological activity, which are further processed by UDP-gluconyl transferases and eventually eliminated from the cell. Adapted from Niederreither and Dolle 2008 [10].

## 3. Embryonic Defects Arising from Dysregulated RA Signalling

*All-trans* retinoic acid (RA) is one of the most important signalling molecules in embryogenesis, required for the development of a large number of tissues. RA homeostasis can be disrupted via maternal diet or genetic/chemical modification to either increase or decrease RA availability and/or signalling relative to normal endogenous levels. This leads to a wide range of developmental defects affecting many tissues. These include antero-posterior axis development (anterior and caudal truncations), CNS abnormalities (posteriorization of the hindbrain), abnormalities of limb, lung kidney and eye development [9,60,62,84,85,86,87,88,89,90,91,92,93,94,95,96,97,98,99,100,101,102,103,104] reviewed [84,85].

The development of the craniofacial/pharyngeal region can also be severely affected by dysregulation of the RA pathway. Observed defects include abnormal development of the pharyngeal arch/artery development, pharyngeal pouch endoderm segmentation and number/migration of neural crest. This produces craniofacial anomalies and cardiac defects affecting the great vessels and outflow tract. Craniofacial anomalies can include anterior pharyngeal arch fusion, defects of the cranial skeleton and teeth, reduction of the frontonasal process, cleft palate/lip, cranial gland, ocular and ear defects. Cardiac defects largely affect the formation/remodelling of the pharyngeal arch arteries into the great vessels and the development of the outflow and inflow tracts by addition of secondary heart field progenitors at the arterial and venous poles of the heart. Neural-crest driven septation of the outflow tract is also abnormal leading to CAT and aorticopulmonary windows [9,76,77,78,79,80,101,105,106,107,108,109], reviewed [10,11,12,13,14,110].

The pharyngeal anomalies and consequent craniofacial and cardiac defects produced by altered RA signalling are very like the abnormalities seen in human 22q11 Deletion Syndrome (22q11DS). This interstitial chromosomal deletion syndrome leads to haploinsufficiency of 1–3 Mb regions of chromosome 22q11 and gives rise to a characteristic phenotype affecting craniofacial, thymus and great vessel/outflow tract development. Animal models for deletion of syntenic chromosomal regions reproduce these anomalies and led to the identification of *TBX1* as the major candidate gene for the 22q11DS. Full and temporal or pharyngeal tissue-specific deletion mouse models for *Tbx1* recapitulate much of the 22q11DS phenotype. Furthermore, ectopic expression of *Raldh2* and down-regulated expression of *Cyp26* genes in *Tbx1* mutant mice indicate that RA levels are up-regulated, likely contributing to the early pharyngeal defects and resulting craniofacial and cardiac defects [69,111,112,113,114,115,116,117,118,119,120,121,122,123,124,125,126,127,128,129,130].

## 4. Retinoic Acid Degradation: The CYP26 Enzymes

The role of the degrading CYP26 enzymes is two-fold. Firstly CYP26-expressing cells act as retinoic acid sinks. Expression of *Raldh2* which synthesises RA and one or more of the *Cyp26* genes is often complementary within a tissue. This establishes a retinoic acid gradient between adjacent high and low RA-expressing regions (Figure 2). This gradient then establishes dose-dependent specific transcriptional readouts, which are important for embryonic patterning. Secondly, expression of CYP26 enzymes protects cells/tissues which are extremely sensitive to RA by preventing inappropriate RA-mediated transcription in these regions.

## 5. CYP26 Structure, Biochemistry and Function

There are three vertebrate *Cyp26* genes, *Cyp26a1*, *b1* and *c1*, which are active during development and later life. These enzymes are cytochrome P450s (CYPs), enzymes named for the 450 nm absorption band of their carbon-monoxide bound form. They are members of the membrane-anchored microsomal (endoplasmic reticulum-ER) p450 mixed function oxidase superfamily, which are important in the metabolism of both endogenous and exogenous biologically active molecules.

All CYP enzymes, including the CYP26s, contain several membrane-spanning domains and a conserved C’-terminal haem-binding domain of amino acids surrounding a central cysteine residue, which binds the iron molecule required for enzymatic activity. This comprises a four-helix (D, E, I and L) bundle, helices J and K, two sets of β sheets, and a coil region. The haem-binding loop contains the conserved consensus sequence Phe-X-X-Gly-X-Arg-X-Cys-X-Gly, which includes the absolutely conserved cysteine required for iron binding and is positioned on the proximal face of the haem just before the L helix. The Glu-X-X-Arg motif in helix K also on the proximal side of heme is also absolutely conserved and is predicted to stabilize the core structure. A final consensus sequence in central part of the I helix, Ala/Gly-Gly-X-Asp/Glu-Thr-Thr/Ser is also characteristic of p450 proteins (Figure 3C).

Substrate recognition and binding sites found near the catalytic site and substrate access region undergo flexible changes upon substrate binding which promote the catalytic reaction. These domains, along with those for targeting membrane-bound proteins and amino-terminal anchoring, have much less sequence conservation than the haem-binding domains where the highest structural conservation is to be found. There is however, a generally high reproducibility of overall topography and structural folding despite other sequence variations mirroring differences in catalytic reaction, electron donors and membrane localisation. Localisation at microsomal/ER membranes is frequently mediated by a proline group (Pro-Pro-X-Pro), which forms a hinge region between a basic residue domain and the hydrophobic domain of the amino-terminal region required for membrane anchoring [131,132,133,134,135,136] (Figure 3).

The haem-binding domain is essential for enzyme function. CYP26 enzymes are class II p450 enzymes, meaning that the catalysis reaction requires 2 electrons per cycle provided from NADPH via *cytochrome p450 oxidoreductase* (*Por, also Cpr, Cypor*), which is the obligate electron donor for the CYP26 enzymes. The proteins link via the ER membrane to which they are both bound. POR has two cofactors: flavin mononucleotide (FM) and flavin adenine dinucleotide (FAD). Electrons move from NADPH to FAD, then to FN which interacts with the CYP26 protein following a conformational change forming a redox chain. The two faces of the POR and CYP26 proteins carry opposite charge and the electron flow is enabled by the protein–protein interaction between them, causing them to act as dipoles with salt bridges forming once oppositely charged amino acids interact [135]. Similar substrate specificity was seen for all three CYP26 proteins, with high catalytic activity for RA, with affinity in the range of Km < 100 nM observed in experiments with COS-1 transfected cells. This Km value for RA is similar to the concentration of RA in a range of tissues and is roughly 1000 times higher than to other CYP enzymes. Enzyme turnover rate was around 1–10 pmol/min/pmol, similar to that of other mammalian p450 enzymes. Much lower levels of substrate specificity were seen with 9-cis-RA, retinaldehyde and other retinoids for CYP26A1 and B1, whereas CYP26C1 was able to catalyse 9-cis-RA oxidation at an equivalent rate to RA. CYP26C1 also cleared RA metabolite 4-oxo-atRA more efficiently than the other two CYP26 enzymes. Overall, the metabolisation of RA by the CYP26 enzymes is up to 10^4^-fold higher than for other CYP enzymes shown to hydroxylate RA in the liver, such as CYP3A4, CYP3A5, CYP3A7, CYP2C8 and CYP2C22 [135,137,138,139,140,141,142,143,144,145,146,147,148,149,150].

In experiments based on microsomal liver extractions and transfected cells, CYP26 enzymes can hydroxylate RA to three major metabolites 4-hydroxy, 4-oxo, 18-hydroxy plus 5, 8 epoxy all-trans RA and more polar products which include dihydroxy, mono-oxo and mono-hydroxy derivatives. These are thought to be produced from the β-ionone ring of RA following multiple hydroxylations. These chemical forms are less biologically active than RA and undergo further glucuronation by UDP–glucuronosyltransferases (e.g., UTP2B7) to 4-*O*-β-glucuronide and are eventually eliminated from the cell (Figure 1) [139,143,144,146,150,151,152].

It has been suggested that some metabolites of RA may still have biological activity within the embryo and some evidence has been put forward for this idea. All three CYP26-generated RA-metabolites can regulate *Cyp26* expression in the chick and each is capable of rescuing the RA-deficient phenotype of the VAD quail embryo [153]. Furthermore, 4-oxo-RA causes anteroposterior defects and the induction of *Hoxb4* and *9* in *Xenopus* embryos, and is a high-affinity activating ligand of RARβ [154]. In zebrafish 4-oxo-RA produces the same range of development abnormalities as all-trans-RA but at a lower efficiency [155]. However, genetic experiments in the mouse suggest that this does not occur in vivo, since the majority of *Cyp26a1* null embryos on a *Raldh2* heterozygous background survive well past birth. If part of the *Cyp26a1* null phenotype is the result of impaired RA-metabolite signalling, then crossing on the *Raldh2* haploinsufficient background should exacerbate the phenotype by decreasing the levels of substrate for the oxidizing enzymes, thus leading to lower levels of RA-metabolites. Additionally, *Cyp26a1* can rescue excess RA phenotypes, whereas if the CYP26 metabolites were active it might be expected to potentiate the effect of excess RA. Therefore defects observed in Cyp26 loss-of-function models are thought to be the result of excess/ectopic RA and not absence of RA-metabolites. It has been suggested that in normal development, although the RA-derived CYP26 metabolites above can be biologically active, they are very rapidly conjugated, mainly as glucuronates, and eliminated by excretion before accumulating to levels at which they can exert a biological effect [153,156]. 

Despite similar biochemical functions, the individual three CYP26s proteins have only approximately 55% amino acid identity to each other, mostly based around the C’-haem domain (Figure 3A). However, each CYP26 protein is highly evolutionarily conserved between species (human, mouse and zebrafish) with the human to mouse amino acid identity being over 80% for each protein and CYP26B1 the most highly conserved and CYP26C1 the least conserved (Figure 3B). In amphioxus, three similar *Cyp26* genes, *Cyp261-3* have been identified which are clustered within the genome and possibly are the result of lineage specific duplication of an ancestral gene. *Cyp261* and *3* are very responsive to RA levels and are induced to protect against teratogenic outcomes of inappropriate RA-mediated transcription. *Cyp262* however, exhibits a complex developmental pattern, and is suggested to mediate developmental patterning in amphioxus [157,158]. An RA-inducible *Cyp26* homologue has also been identified in ascidians expressed in the neural plate and tail-bud in a similar fashion to *Cyp26a1* [141,146,159,160,161,162,163], reviewed [137,139,144,164].

Red letters: small hydrophobic residues, blue letters: acidic residues, magenta letters; basic residues, green letters: hydroxyl, sulfhydryl, amine, and glycine residues. Grey asterisk: single fully conserved residue, grey colon: conservation between residues of strongly similar properties, grey period: conservation between residues of weakly similar properties.

The individual *Cyp26* genes have differing domains of expression within the developing embryo and in adults. Generally, expression of CYP26 is found in those tissues with particular sensitivity to RA (e.g., the tail-bud, limb, hindbrain and pharyngeal regions) and is often found in complementary domains to the RA-synthesising RALDH enzymes (Figure 2). There is some variation in the embryonic expression of each specific gene between different vertebrate species at specific developmental stages (mouse, rat, chick, zebrafish and Xenopus). However, the combined domain of expression of all three *Cyp26* genes considered as a whole, is overall similar between the species. An excellent overview of the expression domain of all three *Cyp26* genes during development in four different species can be found in White and Schilling 2008 [143,159,160,161,165,166,167,168,169].

Conclusions drawn from studies on the effect of expression of CYP26A1 have suggested that expression within a cell can deplete it completely of RA. As described above and in Figure 2, this allows CYP26-expressing tissues to act as RA sinks. When combined with adjacent RA-synthesising expression domains, this allows the generation of RA gradients across a tissue.

Furthermore, expression of CYP26 enzymes within specific tissues or cell types allows protection of these selected cells/tissues from RA signalling even when the overall RA availability across a region is high. This allows the CYP26-positive cells to adopt different cell fates compared unprotected neighbouring cells exposed to higher RA levels [103,151,169,170,171,172].

As yet, further studies examining the effect of relative levels of expression of RALDHs and CYP26s within the same cells/tissues on RA levels and transcriptional outcomes have not been undertaken in a systematic fashion. However, a study investigating complementary and cellular co-expression of RALDH and CYP26 enzymes in different regions of the adult human brain have been suggested to underlie differential paracrine and autocrine RA signalling respectively, although resulting functional differences remain to be explored [173]. 

This complex and dynamic expression of RALDH and CYP26 enzymes throughout development has been shown to be an important mechanism for regulating RA availability and thus RA autocrine and paracrine signalling and subsequent RA-mediated transcriptional outcomes during development and regeneration. These mechanisms are discussed in detail in a variety of tissue contexts in the sections below.

Expression of both *Cyp26a1* and *Cyp26b1* mRNA can be induced by exogenous RA in a dose-dependent manner in developmental and adult tissues. There is a clear response of both hepatic *Cyp26* mRNA and total retinol concentration to both dietary Vitamin A and exogenous administration of RA, with the latter being a particularly strong response. In response to low, adequate and increased levels of dietary Vitamin A, rats showed dose-dependent increases in *Cyp26a1*. A study in which similar levels of dietary Vitamin A were fed to rats across their lifetime gave similar results at every time point examined, from young to old age [174,175]. In response to exogenous RA in Vitamin A-deficient rats, *Cyp26a1* mRNA in the liver increased over 6 h to 2000-fold, and then decreased to baseline by 72 h. In other tissues including testis, lung, kidney and small intestine, RA treatment also led to increased *Cyp26a1*, albeit at lower levels with a 10-fold increase after 10 h [139,144,176,177,178]. Explants from embryonic E8.5 tail bud which expresses *Cyp26a1* and somites, which do not, showed rapid upregulation of RA-induced *RARE-lacz* reporter levels 3.5 h after exogenous at RA treatment. In tail bud explants *RARE-lacz* reporter levels were reduced back to baseline by 6 h after treatment whereas somatic explants continued to show high levels of atRA-reporter expression. Ectopic *RARE-lacz* expression in tailbud and pharyngeal regions also shows increases levels of RA within *Cyp26a1^−/−^* embryos. Furthermore, RA bead explant experiments/treatment with RA/RAR agonists in zebrafish, chick and Xenopus embryos induce local expression of *Cyp26a1* [103,153,179,180,181,182]. This response appears to be mediated co-operatively by the three and a half RARE sites found within the 2.2 kb upstream sequence of the *Cyp26a1* transcriptional start site (TSS), one close to the TSS and the remaining sites 2 kb more distal [178,183,184,185]. Additional more distant RARE sites, positioned more than 10 kb away from the *Cyp26a1* TSS and conserved between six species including mouse and human which were further validated for functionality have also been identified using an in silico approach [186]. Obviously, other transcription factors will also act to regulate *Cyp26a1* expression in addition to RA binding. For example, a known proximal SP1/SP3 binding site has been found to increase *Cyp26a1* promoter activity in response to RA in co-operation with the nearby RARE binding site [183] and it has been shown that HOXA10V2 enhances induction of *CYP26A1* by RARα/RXR primed transcription in NB4 cells [187]. Epigenetic factors such as removal of repressive complexes, including PcG proteins (eg Suz12) and HDACs, recruitment of activators (e.g., pCIP/p300) and PolII, H3K27 acetylation, H3K9 methylation, differential chromatin regulation by RARβ2 are all also linked to regulation of *Cyp26a1* expression in different contexts [188,189,190,191].

The response of *Cyp26b1* mRNA to RA in the liver is less than that of *Cyp26a1*. As for *Cyp26a1*, *Cyp26b1* mRNA levels increased in a dose-dependent linear fashion, but overall up-regulation was not more than 10-5-fold and again reduction to baseline levels was accomplished by 72 h [144,192]. In the neonatal rat lung when treated with Vitamin A/RA/*Cyp26b1*, mRNA increased at a rapid rate to a higher more persistent level than *Cyp26a1* mRNA [193]. *Cyp26b1* has also been reported to be induced in response to RA in aortic smooth muscle cells [194] and in response to local RA-bead implantation in chick embryos [153]. In naïve CD4+ T cells, *Cyp26b1* is induced by 1–10 nm RA and this up-regulation was inhibited by TGFβ1, TGFβ2 and IL-12 [195]. TGFβ family member activin has also been shown to inhibit *Cyp26b1* in telencephalic neural precursors and mouse granulosa cells [196,197]. Previously consensus RARE sites have not been reported within the *Cyp26b1* promoter, however the in silico study above generated a number of DR5 RARE binding sites for both mouse and human *Cyp26b1* [186]. However, these have not yet been functionally validated. In TM3 cells as a model for gonadal somatic cells *Cyp26b1* has also been shown to be RA-independently activated by SOX9 and SF1-expressing constructs and repressed by FOXL2. In *Sox9/Sf1-* and *Foxl2-*deficient gonads, *Cyp26b1* transcription was respectively decreased and increased approximately 20-fold compared to wild-type [198]. All three *Cyp26* genes may be up-regulated by TBX1 as expression is down-regulated in *Tbx1* null embryos. It is unknown if this is a direct or indirect transcriptional effect although putative conserved TBX1 binding sites have been found in *Cyp26b1* upstream sequences (C. Roberts, P. Ataliotis unpublished data). Furthermore, activation of *Cyp26b1* expression via PPAR agonists, presumably via promiscuous binding of PPAR receptors to RAREs have previously been reported [199]. *Cyp26c1* has also been reported as being both up and down regulated by RA [141,153] and in silico RARE sites have been predicted [186] but little information exists about other possible regulators of the *Cyp26c1* promoter. *Cyp26c1* is on the same chromosome as *Cyp26a1* within 13 kb distance of each other, raising the possibility that there may be regulatory elements in common [141,161].

The function of the CYP26 enzymes can be chemically blocked by a variety of retinoic acid metabolism blocking agents (RAMBAs). These have been developed with possible clinical applications in mind: increasing RA levels by inhibiting CYP26 function could provide a more specific less toxic approach to RA treatment for a variety of conditions including cancer and dermatological diseases. The best known of these are R115866 (talarozole), R116010, ketocozanole and liarozole. These compounds, particularly R116010 and R115866, have also been useful in exploring CYP26 biological function in variety of animal models in vivo and in vitro. R115866 and R116010 contain imidazole and triazole and are more potent inhibitors of CYP26s with an IC_50_ of 4–5 nM each compared to liarozole and ketoconazole with IC_50′_s of 2100 and 550 nM, respectively. Inhibition of CYP26B1 has also been demonstrated for these molecules. While no compound is totally specific, the selectivity for CYP26 in particular, is evidenced by trivial inhibition of other CYP-dependent synthesis of estradiol and testosterone (micromolar concentrations of R115866/R1160101 are required to inhibit CYP19, CYP 17, CYP 2C11, CYP3A and CYP2A1, CYP2B1/1). 

Oral administration of R115866 to adult rats resulted in raised levels of RA in plasma, skin, fat, kidney and testis. R115866 also reproduces known retinoidal effects including vaginal keritanization, induction of epidermal hyperplasia and epidermal transformation and up-regulation of *Cyp26* mRNA expression in rat liver. These effects can all be reversed by administration of retinoic acid receptor antagonists suggesting that R115866 inhibition of CYP26s results in an increased availability of endogenous RA and exposure of developing embryos to R115866 results in a range of developmental defects which phenocopy those of exogenous RA. Further compounds based on these four initial molecules are still in development [126,200,201,202,203,204], reviewed [205]. 

## 6. CYP26 in Human Genetic Disease

Various polymorphisms potentially affecting CYP26 activity have been reported in humans, along with some disease-causing mutations (see Table 1 for details) with additional silent polymorphic changes also reported (reviewed [139,149]). Polymorphism screens of *CYP26A1* from healthy individuals of varying ethnicity identified at least 13 single nucleotide change polymorphisms (SNP). Reduced enzymatic functionality was confirmed for F186L, and C358R which were found to have reduced RA metabolising activity of between 40% and 80% compared to wild-type. Further silent changes have been reported for both *CYP26A1* and *CYP26B1* [206,207]. A further single nucleotide deletion polymorphism g.3116delT (premature stop) associated with spina bifida was also found to attenuate *CYP26A1* RA metabolising activity [208]. Polymorphisms in both *CYP26A1* (rs4411227 C/G genotype or C/C+C/G compared to G/G) and *CYP26B1* (rs9309462 C/T genotype and C allele, rs138478634 G/A change in exon 5) alone or combined polymorphisms (*CYP26A1* rs4411227 and *CYP26B1* rs3768647/rs930946) have an increased risk of oral and pharyngeal cancer [209,210,211,212]. Raised levels of CYP26 enzymes have also been reported in a range of other cancers including breast, colorectal and head/neck cancers, reviewed [149]. Haploinsufficient microdeletions encompassing the CYP26A1 and CYP26C1-containing chromosomal region has also been reported with differing phenotypes of either optic nerve aplasia or premature skeletal and dental aging combined with retinal scarring and autism [213,214].

A variety of missense variants in *CYP26B1* shown to abrogate protein function are associated with a range of human neural tube defects, craniofacial and skeletal defects and defects affecting other tissues including the heart (atrial septal defects and dextrocardia) [207]. An alternatively spliced *CYP26B1* lacking exon 2 and has reduced metabolic functionality of ~30% compared to full length *CYP26B1* and is expressed at higher levels in vascular cells atherosclerotic lesions. This expression was further upregulated with exposure to RA. A polymorphic variation rs2241057C/C (minor allele) L264S is associated with larger macrophage-positive atherosclerotic lesions, whereas carries of the major allele rs2241057T/T have an increased risk of Crohn’s disease, an autoimmune condition of the gut. Additionally, *Cyp26b1* is upregulated in atheroschlerotic lesions and associated there with activated macrophages, suggesting that Cyp26b1 may be important in RA clearance in the arterial wall [215,216,217]. 

Several papers have reported human skeletal abnormalities in patients with *CYP26B1* mutations. Three siblings carried a homozygous CYP26B1 c.1088G > T transversion of *CYP26B1*, predicting a p.Arg363Leu substitution (one died in utero at 35 weeks and two terminations). All three displayed severe skeletal abnormalities including craniofacial malformations, radio-humeral fusions, hypoplastic pelvis, calvarial plate hypoplasia and mineralisation defects, oligo and arachnodactyly, narrow thorax, premature bone maturation and occipital encephalocele. The mutation affected the conserved core-stabilising K-helix region required for enzymatic catalysis and produced a similar reduction in ability to metabolise RA to that of a null truncation [202]. Furthermore, predicted loss of function mutations in *CYP26B1* have produced skeletal phenotypes analogous to that of Antley–Bixler and Pfeiffer Syndromes as described in Table 1 [202,218]. Microdeletions of chromosome 2p13.2–13.3 leading to haploinsufficiency of *CYP26B1* and *EXOC6B* have been associated with phenotypes of intellectual disability, language delay, hyperactivity, dysmorphic facies and vertebral and/or craniofacial abnormalities [219].

Additionally, a variety of homozygous or compound heterozygous mutations, including missense, nonsense, frameshift, splicing and exon deletions in *POR*, the CYP26 obligate electron donor give rise to abnormal steroidal profiles with and without Antley–Bixler skeletal phenotypes [220,221,222].

Missense mutations in *CYP26C1* which reduce enzymatic metabolism of RA act as modifiers of missense SHOX mutations conferring the more severe skeletal phenotype of short stature and limb defects (shortening and bowing of the radius together with distal hypoplasia of the ulna and mesomelia) compared to phenotypically normal/mild short stature in family members carrying only the SHOX mutation [223]. It has since been shown that missense mutations and splice variants of *CYP26C1* causing reduced enzymatic activity [224]. Homozygote 7 bp duplications leading to a frameshift and a premature stop c.844_851dupCCATGCA p.Glu284*fs*X128 and/or compound heterozygote mutations of the duplication with an inactivating missense mutation c.1433G > A p.Arg478His give rise to focal facial dermal dysplasia, where an abnormal epidermis and replacement of the dermis by connective tissue and loss of subcutaneous tissues gives rise to skin lesions at the sites of facial fusion during development [225]. Thus, in humans disease-causing variations in the *Cyp26* genes give rise to variety of conditions including neural tube, cardiac and skeletal defects amongst others. This correlates with the phenotypes generated from null and conditional mutations of the *Cyp26* genes in animal models as discussed in more detail below.

## 7. Function of CYP26 Enzymes during Development

### 7.1. Deletion of All Three Cyp26 Genes/Por

In mutant mice lacking all three *Cyp26* genes, *Nodal* expression is ectopically activated in the entire epiblast during gastrulation, via an RARE in the *Nodal* autoregulatory enhancer. This ectopic activation results body axis duplication in approximately half the embryos and severe patterning defects in the brain in the remaining embryos, similar to those seen in *Cyp26a1/c1* null embryos. About 25% of *Cyp26a1/c1^−/−^* embryos also displayed duplication of the primitive streak whereas *Cyp26a1/b1^−/−^* mutants did not, suggesting that *Cyp26a1* extra-embryonic expression is the main *Cyp26* gene in early development, acting non-cell autonomously to keep the epiblast RA-free. However, since *Cyp26a1* null mutants have no gastrulation defects, the other two genes must be able to compensate for its loss. Increased dietary RA was able to increase the frequency of severe phenotypes in triple and double homozygous mutants. These defects could be induced by very high levels of added RA in wild-type diets. This suggests that mutant phenotypic variation may be due to varying maternal RA levels and again highlights the importance of CYP26s in development, as under normal conditions the embryo should be protected against levels of variation of dietary maternal RA [226].

Studies in the chick embryo using the CYP26 inhibitor R115866 bypassed gastrulation stages to look at the effects of CYP26 inhibition when R115866 was added in ovo at stage 10 (10 somite) and stage 14 (22 somite E9.0) and examined following 24/48 h further incubation. Embryos given relatively high doses of R115866 displayed phenotypes similar to higher doses of RA and did not survive past E5 at the very latest. Defects included decreased head mesenchyme, smaller otic vesicles and loss of anterior tissues such as the forebrain. Pharyngeal defects included loss of caudal pharyngeal arches/arch arteries, pharyngeal endodermal pouch segmentation and reduced pharyngeal arch artery vascular smooth muscle, plus loss of migrating neural crest and mis-patterned cranial ganglia. Heart defects comprised shorter outflow tracts, abnormal looping and pericardial oedema. Embryos given lower doses of R115866 survived up to E8 and displayed mature cardiovascular defects resulting from abnormal pharyngeal arch artery development including common arterial trunk, double outlet right ventricle and ventricular septal defects. Upregulated retinoic acid levels were confirmed by anterior ectopic shifts of *Hoxb1*. These defects strongly phenocopy those seen in *Tbx1* mutant mouse models for human 22q11 Deletion Syndrome which display dysregulated retinoic acid levels, likely driven by upregulated *Raldh2* and downregulated *Cyp26* expression [65,69,126,127,227].

Knockout mice for *cytochrome p450 oxidoreductase* (*Por*), which is the obligate electron donor for the CYP26 enzymes and necessary for their function, display very severe phenotypes. These included growth retardation, axial rotation defects abnormal head and caudal development, open neural tubes, pharyngeal and cardiovascular defects, which are lethal by E10.5. These embryos exhibit ectopic RA-signalling and can be partially rescued by genetic down-regulation of RA using the *Raldh2* null allele [228,229,230], thus displaying the importance of CYP26 function during development.

### 7.2. Loss-of Function Cyp26a1 Models

Targeted disruption of *Cyp26a1* and *Cyp26b1* leads to phenotypes that are similar to the application of exogenous RA. Two separate knock-out mice for *Cyp26a1* have been found to have a phenotype which includes posterior truncations and sirenomelia, abnormal development of the posterior gut and urogenital system, homeotic posterior transformations of the vertebrae and hindbrain and cranial nerve patterning defects [179,231]. These anomalies are accompanied by an increase in RA signalling which has been linked to the mis-patterning of the brain and vertebrae and caudal truncation associated with down-regulation of *Brachyury* and *Wnt3a* [103,232,233,234,235].

Caudal truncation malformations appear to be mediated by inappropriate activation of *Rarg* (retinoic acid receptor gamma). Loss of *Rarg* confers resistance to the caudal abnormalities induced by teratogenic RA doses and rescues *Cyp26a1^−/−^* caudal regression/lethality by restoring normal expression of tail bud genes including *Wnt3a*, *Brachyury* and *Fgf8* [81,103,236]. In the wild-type state it seems *Cyp26a1* function is to protect against the effects of excess environmental RA, particularly since RA can induce *Cyp26a1* expression via RARE sites in its promoter, although this system is not sufficient to degrade teratogenic doses of exogenous RA [103,142,183,237].

In some embryos developmentally arrested between E8.5 and 9.5 pericardial oedema, cardiac looping and dilation defects were also observed. As described above, forebrain defects of varying severity could be induced in *Cyp26a1^−/−^* embryos when given subteratogenic doses of RA that do not result in defects in wild-type embryos. The severity of the defect varied in dose and time-dependent fashion: an earlier period of administration at a low dose of RA (E6.5–7.5 20ugRA/g food) led to E9.5 lethality, a phenotype possibly linked to the gastrulation defects observed in triple-deleted at *Cyp26a1/c1^−/−^* embryos. Later RA-supplementation (E7.5–8.5) led to anterior truncations which could affect all rostral brain regions at E9.5 (high dose) to lesser truncations affecting only more anterior regions (lower dose). Blood vessel endothelium of the yolk sac and within the embryo was also affected with much lesser plexus development observed [238].

Similar phenotypes are also observed in the zebrafish *cyp26a1* null mutant, giraffe [239]. *Cyp26c1* knockouts alone seem to have no discernible embryological defects. This may be due to functional redundancy with *Cyp26a1* with which it is expressed in an overlapping pattern and is found on the same chromosome [161,240]. In addition to the subset with gastrulation axis defects described above, double homozygous mutants for *Cyp26a1/c1* display a more severe RA embryopathy phenotype than either mutant alone, with lethality by E11.0. This includes CNS patterning abnormalities, a reduced size of the head, eye, frontonasal region and an open neural tube between the fore and hindbrain, hypoplastic PA1 and 2 and abnormal NCC migration. The NCC defects can be rescued in the context of a *Raldh2^−/−^* genetic background, again suggesting that CYP26s have a protective role against inappropriate RA exposure during development. This idea is supported by the studies described above where *Cyp26a1^−/−^* RA-supplemented embryos exhibit more severe phenotypes when exposed to maternally administered subteratogenic doses of RA that do not result in defects in wild-type embryos [226,240,241].

### 7.3. Loss-of Function Cyp26b1 Models

*Cyp26b1* null mutants have been reported to have severe meromelia-like limb defects with oligodactyly, micronathia and lethality immediately after birth as a result of respiratory distress. Abnormal limbs undergo increased apoptosis and become proximilized and ectopic distal RARE-lacZ expression was observed. The apoptotic defect appeared to be mediated via *Rarg* but not the proximo-distal patterning anomalies [242,243,244,245]. These defects are discussed in more detail in the sections below.

Two *cyp26b1* mutants in the zebrafish, *stocksteif* and *dolphin* exhibit a reduction in midline cartilage of the neurocranium and pharyngeal arches and severe over-ossification of the axial skeleton and craniofacial bones leading to fusion of the vertebrae, which can be phenocopied in mouse embryos by treatment with CYP26 inhibitor R115866. As discussed above, *CYP26B1* mutations in humans also confer skeletal defects [202,203]. 

Craniofacial defects are also reported in the mouse, including cleft palate, reduced or absent incisor development, micrognathia and absent posterior nasopharynx [246]. Further investigation of the cleft palate defect revealed that *Cyp26b1* null embryos have a 100% penetrant cleft palate defect, arising from reduced proliferation in the bend region of the palatal shelves [246,247]. Craniofacial ossification is reported to be reduced, along with severe abnormalities in the craniofacial skeleton where many bones are missing or deformed with abnormal fusions. Similar malformations are observed in the trachea, larynx, auditory system and dental development. Molecular markers reveal hindbrain patterning to be relatively normal but disturbance of caudal neural crest migration and cranial nerve patterning is present [246], matching reports using a *Cyp26b1* morpholino (MO) in which cranial nerve patterning and *Dlx2* expression in the neural crest is down-regulated in zebrafish morphants [248].

These loss-of-function experiments reveal a wide range of important roles for the *Cyp26* genes in regulating RA availability during development, from gastrulation to whole embryo anterior–posterior patterning to organogenesis. Further studies which focus on the role of the Cyp26s in specific tissues/organs have elucidated particular cellular mechanisms requiring regulation RA by Cyp26s during development and have explored the relative importance of these enzymes in acting to secure RA-negative cells/tissues versus their role in establishing RA gradients. These reports are discussed in detail below for several developing orgains/tissues.

## 8. Forebrain

Studies in avian and mouse embryos indicate RA synthesis and signalling are required for forebrain development. Loss of RA causes increased apoptosis and decreased in anterior tissues which has been linked to dysregulation of signalling molecules including FGF8, WNT and SHH. It is tempting to speculate that these molecules, in conjunction with RA levels mediated by RALDH2 and CYP26A1, act in similar fashion in the forebrain as in the hindbrain (see below) to contribute to anterior patterning.

In mouse mutants, deletion of *Cyp26a1* alone does not produce significant anterior defects. However, this is likely to be the result of redundancy with *Cyp26c1*, as double homozygote knockout embryos display severe anterior defects early in development [226]. Anterior truncations are also seen in *Cyp26a1* mutants when dosed with exogenous RA in concentrations which are aphenotypic in wildtype embryos [238]. Studies in zebrafish and chick give similar results. Forebrain a/hypoplasia is seen in chick embryos treated at the 10-somite stage with CYP26 inhibitor R115866. In zebrafish *cyp26a1* mutants, posteriorisation of the entire neural plate is seen in response to non-teratogenic doses of RA and smaller heads are seen in *cyp26a1/c1* double morphants and *giraffe* (*cyp26a1*) mutant embryos injected with *cyp26c1* morpholino. Moreover, expression of *cyp26a1* in the anterior neural plate seems to be both RA dependent and independent. Regions of lower *cyp26a1* expression upregulate this expression via RAREs when exposed to RA. However, high levels of *cyp26a1* can also be induced in an RA-independent fashion [180,249,250,251]. An RA-independent SOX: OCT-binding motif in the *cyp26a1* promoter was found to drive *cyp26a1* expression in the anterior neural plate of the zebrafish and *cyp26a1* expression was lost in *sox2/3/19a/19b* quadruple morphants. Finally, ChIP experiments were positive for SOX at the *cyp26a1* promoter indicating a direct interaction [252,253]. Transcription factors *tgif* and *zic1* are also reported to be involved in the initiation and maintenance of anterior neuroectodermal *cyp26a1* expression. *Zic1* knockdown caused severe forebrain and midline defects. These morphants displayed increased levels of RA resulting from decreased *cyp26a1* expression in the forebrain (but not the hindbrain) and repression of dorsal *bmp* signalling [254].

*Tgif* is a TALE class homeobox transcription factor. *Tgif* loss of function produces severe anterior neural tube and midline defects in human [255,256,257,258,259,260,261], zebrafish and some mouse mutants [262,263,264,265,266]. TGIFs can bind to RXR/RAR complexes and in the zebrafish loss of *tgif* results in reduced anterior *cyp26a1* and *raldh2* expression at gastrulation [262,266,267]. Loss of *tgif* function phenocopies those of *cyp26a1* mutants/morphants and ectopic *tgif* expression is sufficient to induce *cyp26a1* [266]. *Raldh2^−/−^* mice also exhibit severe forebrain hypoplasia phenotypes, and the severity of these anterior defects increases with the additional removal of *Raldh3*. Together, the *Cyp26a1* and *Raldh2/3* data indicate control of RA availability is crucial for proper forebrain development [268,269,270]. 

A later role for RA signalling exquisitely regulated by enzymatic control of RA availability has been demonstrated in the development of the chick dorsal forebrain. A central region of *Raldh2*-expressing dorsal mesenchyme overlying the forebrain roof-plate flanked by *Cyp26a1/c1* expressing mesenchyme gives rise to a central region of anterior–posterior low-high RA activity with low proliferation in the dorsal forebrain. Experimental manipulation of these regions by electroporation of VP16 or dominant-negative RA receptor constructs suggested that *Cyp26a1* expression was required to restrict RA-signalling which conferred dorsal forebrain identity and was required for invagination of Rathke’s Pouch. Furthermore, degradation of RA by *Cyp26a1* in the flanking regions to provide an RA-negative domain was necessary for the development of the choroid plexus [271].

## 9. Hindbrain Patterning

Retinoic acid signalling is known to be required for neuroectodermal anterior–posterior patterning as part of a complex signalling network also encompassing roles for FGF and WNT signalling. The role of RA in patterning the hindbrain into segments known as rhombomeres has received particular attention, especially with regard to the role of the *Hox* genes as direct RA targets genes. Each rhombomere has a specific anterior–posterior identity conferred by combinatorial transcription factors such as *Hox* gene expression and this identity is important in a number of further developmental events including brain and otic development and neural crest migration into pharyngeal arches (reviewed [272,273,274,275,276,277,278,279,280,281]).

The RALDH2 synthesising enzyme is expressed only in presomitic mesoderm/somites and not in the neuroepthelium, so atRA signalling in the developing neuroectoderm is of a paracrine nature. It can be hypothesised that RA gradients might be generated in the developing nervous system based on the distance RA must diffuse from generating RALDH2 tissues. This model can be amended to also consider CYP26 RA-degrading expression within the neuroectoderm.

CYP26s are expressed at high levels in very dynamic patterns during nervous system development. *Cyp26a1* is expressed first and most anteriorly beginning with transient expression in r2 at E8.5 in mouse with a posterior r2/r3 boundary which shifts posteriorly later to r4/r5. *Cyp26b1* is expressed strongly in r5 and more weakly in r3 at E8.0, but by E9.5 broader expression is seen, with strong expression at r5/r6 and weaker expression domain extending up to r2. *Cyp26c1* expression is initially in r2 and r4 and then restricted to r2. In zebrafish, *cyp26a1* is first seen with a posterior boundary at r3/r4 at 8.5hpf which then moves anteriorly to r2/r3 by 10hpf and more anteriorly still at 11hpf. *Cyp26b1* expression begins later at 10hpf in r3/r4 and then expands up to r2 by 15hpf as development proceeds, as for the mouse. *Cyp26c1* expression overlaps with, but precedes, *cyp26b1* expression at each stage. 

Various types of loss-of-function models produce hindbrain patterning defects. In mouse embryos, loss of *Cyp26a1* produces a partial r3 to r4 transformation accompanied by an enlargement or r4 [179,231]. When both *Cyp26a1/c1* are deleted, the presumptive r1–r4 region is posteriorised and loses segmentation [226,246]. Hindbrain abnormalities have not been reported for individual deleted alleles of *Cyp26b1* and *Cyp26c1*. 

In the zebrafish, *cyp26a1* null embryo r1–r3 is slightly reduced in size and r4 slightly enlarged. Most caudal hindbrain region r7/r8 increases in length [239]. Alone, knock-down of *cyp26b1* and *cyp26c1* or both, has minimal phenotype beyond a very small shortening of the hindbrain. However, depletion of either *cyp26b1* or *cyp26c1* in a *cyp26a1* null context strongly exacerbated the *cyp26a1* phenotype. C*yp26a1/b1 depletion causes further expansion of* r4 and there is a shift of the r6–r7 boundary towards r5 whereas depletion of *cyp26a1/c1* leads to hypo or aplasia of r3, with expansion of r4 up to the presumptive cerebellar region. There is a similar shift of the r6/r7 boundary towards a slightly reduced r5. When all three *cyp26* genes are removed r3 and r5 are lost completely and r4 lies next to the cerebellum, with r5/r6/r7 boundaries all shifted towards the cerebellum [250].

Differing experimental models in the zebrafish and mouse have led to differing interpretations of these possibilities for hindbrain patterning. A *rare;yfp* transgenic zebrafish demonstrated an RA-response gradient at the hindbrain–spinal cord junction with little RA signalling evident anteriorly, increasing to obvious expression in the posterior hindbrain and spinal cord. Loss of RA production led to loss of the rare-driven signal, which could be restored by re-establishing zones of RA-production in the somatic mesoderm [180]. More recently, genetically coded probes for RA (GEPRAs) have been used to confirm RA gradients consistent with this previous work in zebrafish [282]. Another zebrafish study suggested that this RA gradient grows increasingly steeper with time, leading to concentration-dependent anterior to posterior specification of rhombomere identity, with RA concentrations necessary to establish posterior identity being reached later in development [283]. Experiments in the mouse linked expression of RA-responsive gene *Hoxb1* expression in r4 to that of *Cyp26* genes suggesting that *Cyp26* expression was linked to establishment of rhombomeric boundaries. Posterior *Cyp26* boundaries specify the anterior boundaries of genes conferring rhombomere identity. In this model, length of time of RA exposure rather than concentration drive hindbrain segmental fate [284]. Similar results from a third investigation in the zebrafish proposed that distinct regions of CYP26-driven degradation of RA over time, combined with changing RA sensitivity, specified increasingly posterior boundaries of RA dependent gene transcription necessary for rhombomere segmentation in a step-wise manner [250]. 

Finally, a mechanism which combined both RA gradient and CYP26 RA degradation in the patterning of the hindbrain was proposed on the basis of bead implantation experiments, and mathematical modelling. This requires FGF and RA to act together in parallel in the neural ectoderm to control *cyp26a1* expression (RA induces and FGF inhibits *cyp26a1*) to generate a robust RA gradient which gives rise to rhombomere positional identity. *Raldh2* in the somatic mesoderm produces RA which diffuses through the neural ectoderm. RA is degraded by Cyp26 enzymes at differing rates giving rise to a gradient across the hindbrain which confers rhombomere identity. Thus, FGF and RA act together to regulate *cyp26* expression leading to the production of a stable yet flexible RA gradient which grows along with the embryonic axis [159,180,249,285,286,287] (Figure 4).

Identity switching and cell intermingling takes place during rhombomere segmentation to establish and maintain homogeneous identity and sharp rhombomere boundaries during hindbrain development. Morphogen (RA) signalling fluctuation initially established rough boundaries of gene expression requiring mutual repression of *hoxb1* and *egr2* to restrict expression to r4 and r3/5 respectively [288,289] whereas Eph-Ephrin driven signalling sharpens borders by preventing intermingling (reviewed by [290,291]). A recent paper has added further insight in to the processes that fine tune identity switching during the hindbrain segmentation process via the community effect. Transplantation experiments showed that cells which intermingle during boundary formation switch their identity to match that of their neighbours, a process mediated by feedback between segment identity as specified by expression of *egr2* (*krox20*) and *cyp26b1* and *cyp26c1*. Lower expression of *cyp26b1/c1* in r3 and r5 is regulated by repression by *egr2*. Cells in neighbouring r2/r4/r6 do not express *egr2* and have higher expression of *cyp26b1/c1*. Thus, RA levels are higher in r3/r5 than in other rhombomeres leading to repression of *hoxb1*. *Egr2* directly autoregulates itself, further repressing *hoxb1*. Higher expression of *cyp26b1/c1* in segments not expressing egr2 (e.g., r4) leads to lower levels of RA thus allowing expression of *hoxb1*. Autoregulation of *hoxb1* expression in turn represses expression of *egr2* in these segments. If cells from r3/r5 intermingle into r4, they are surrounded by cells expressing *cyp26b1/c1* at higher levels which leads to a reduction of RA within the intruding cell as only paracrine non-cell autonomous RA signalling is active in the neuroectoderm. Lower concentrations of RA then repress *egr2* expression within ectopic r3 cells via the stimulation of *hoxb1* expression which represses *egr2*. This then causes the r3 cell to change identity to match that of its r4 neighbours. When the ability to regulate *cyp26b1/c1* levels is removed by experimental knockdown, this entire mechanistic chain is prevented [292,293,294] (Figure 4).

## 10. Neuromesodermal Progenitors (NMPs) and Axis Extension

Axis elongation in the developing embryo is driven by the generation of trunk mesoderm and spinal cord neurectodermal tissue. Previously, the generation of opposing RA/FGF-WNT gradients has been linked to the formation and patterning of caudal mesoderm/somitogenesis and spinal cord (reviewed [11,295,296]). As part of this process, *Cyp26a1* expression from gastrulation onwards degrades RA from the most posterior regions, allowing expression of FGF/WNT in a posterior to anterior gradient. An opposing gradient of RA generated by expression of *Raldh2* in somites and anterior presomitic mesoderm and transiently in the node and primitive streak inhibits FGF-driven mesoderm production and allows the drive towards a neural fate, thus maintaining a balance between caudal mesoderm and neural plate. SHH signalling is also implicated in this process, with RA negatively regulating *Gli2*, which is required to promote SHH signalling. Furthermore, loss of *Cyp26a1* and increased RA signalling are apparent in the *Shh* LOF mutants along with caudal truncations similar to those seen in *Cyp26a1^−/−^* embryos, implying posterior *Cyp26a1* expression may also promote GLI2-mediated processing of SHH signalling [229,297,298]. 

It has been shown that these tissues are generated in a progressively more posterior fashion by proliferation and subsequent differentiation of neuromesodermal progenitors (NMPs) found at the caudal end of the embryo. To ensure the embryo reaches the correct length, careful regulation of number of progenitors induced, rate of self-maintenance/renewal and differentiation towards neural or mesodermal fates is necessary [299,300,301,302,303].

NMPs are located in regions expressing WNT and FGF ligands including the border of the node/primitive streak, the caudolateral epiblast and the chordoneural hinge region. Caudal truncations result from interference with either of these signalling pathways as a result of NMP depletion [304,305,306,307,308,309]. Similar truncations/reduced size of caudal structures are reported when RA signalling is deregulated by either deletion of *Cyp26a1* [179,231,239] or *Raldh2* [11,60,229,310]. 

Both FGF and WNT are linked to roles in progenitor maintenance and act as posteriorising signals via *Cdx* genes, which in turn regulate more posterior *Hox* gene expression. Studies in mouse and zebrafish show that WNT signalling is also coupled to mesodermal differentiation of NMPs via an autoregulatory loop, which maintains a mesodermal progenitor niche in which T-box gene *Brachyury* directly regulates expression of *Cyp26a1* in the caudal tail bud, generating a low RA region which allows the expression of *Wnt3a* and *Fgf8* that is inhibited by high levels of RA [232,233,311,312,313]. Furthermore, this autoregulatory loop is regulated by the expression of *Cdx* genes [314,315,316]. Complex interaction and regulation of transcription factors such as *T/Brachyury*, *Cdx*, *Sox2*, *Msgn1* and *Tbx6* with signalling pathways WNT, FGF and RA seem to regulate induction and maintenance of NMPs, mesodermal or neurectoderm fate and embryonic axis elongation [302,308,309,317,318,319,320,321,322,323,324,325].

The complexity of the interacting events with overlapping functions and multiple feedback loops of the pathways involved has made it difficult to assess exactly which functions are mediated by which molecules. Recently, investigators have taken advantage of the iPSC/ESC model where NMPs can be generated outside the complex environment of the embryo [326,327,328,329] to attempt to build a definitive model for these events. This approach combined with RNA-Seq single cell analysis has confirmed much of what had been elucidated from embryonic studies, to wit, a transcriptional network comprised of the *Cdx*, *T/Bra*, *Sox2*, *Msgn1* and *Tbx6* which integrates signals from WNT and RA to control the NMP state. Tight control of RA via *Raldh2/Cyp26a1* expression is important, firstly for NMP induction via *Sox2* and secondly to increase RA levels to drive neural differentiation. *Cdx* genes maintain *T/Bra* expression and thus allow RA degradation via *Cyp26a1* and mesoderm induction as well as acting to posteriorise expressing cells. Cell-autonomous repression of *T/Bra* and *Sox2* by *Tbx6* and *Msgn1* control time and fate of NMP differentiation, promoting mesodermal differentiation and *Raldh2* expression and thus RA signalling. This leads to elevated levels of RA which signal in paracrine fashion to drive neural differentiation, thus balancing neural versus mesodermal fates [330,331,332] (Figure 5). 

Embryonic data including *Raldh2*, *Cyp26a1* and *RARγ* expression and loss-of-function data suggested an early role for RA signalling in NMP induction and anterior–posterior axis elongation but were confounded by a possible maternal contribution of *Raldh2* [60,72,78,79,81,103,179,182,229,231,236,284,333]. Data from single cell analyses confirmed RARγ NMP expression in vivo and in vitro, and showed *Cyp26a1* was expressed at E9.5 but not E8.5. Upregulation of *Cyp26a1* at E9.0 is thought to degrade RA levels to allow NMP maintenance. NMP identity was lost in vitro if RA was removed at differentiation day 0, along with loss of *Sox2* and increased expression of mesodermal genes. On the other hand, exposure to high levels of RA in differentiating NMPs led to gain of pre-neural tube fate at the expense of mesoderm [330] (Figure 5).

Combined with embryonic studies, this suggests a model of the posterior of the elongating embryo where an anterior–posterior gradient of RA is established from RA-synthesis in the caudolateral epiblast plus paracrine RA from more anterior mesoderm counterbalanced by CYP26A1-mediated RA degradation in the posterior tailbud. This gradient controls the birth and anterior–posterior position of different trunk progenitor cells. Posteriorly, CYP26A1-mediated inhibition of RA plus FGF/WNT signalling induces *Bra/Tbx6*-positive mesodermal tissue from NMPs. Loss of *Cyp26a1* expression causes loss of *Sox2*, loss of NMP identity and upregulation of mesodermal specific genes. Anteriorly high RA promotes induction of *Sox2/Nkx2.1* pre-neural fates from NMPs. NMPs themselves (*Bra/Sox2*-positive) require low levels of RA for induction and maintenance [307,330] (Figure 5).

Thus, in this model, the relative levels of mutually antagonistic RA and WNT signalling control *T/Bra* and *Sox2* expression and regulate the switch between mesodermal and neural differentiation and therefore the proportion of the two cell fates. Experimentally altering RA/WNT levels in differentiating NMPs can alter the relative proportions of mesodermal (*Tbx6*+) and neural (*Sox2*+) cells produced. Moreover, this model allows an RA-driven feedback loop to maintain the proper balance between mesoderm and neural fates. RA signalling from differentiated mesoderm induces *Sox2* in undifferentiated NMPs, thus promoting neural differentiation. Alternatively, reduced formation of mesoderm plus *Cyp26a1* expression reduces RA thus decreasing *Sox2* expression in NMPs inducing a switch towards mesoderm production. This establishes an equilibrium in the direction of differentiation towards neural or mesodermal tissue in the bipotential NMP progenitors which expedites correct axis elongation [330] (Figure 5).

## 11. Skeletal Tissues: Craniofacial, Axis and Limb

Deletion of *Cyp26b1* revealed a very severe limb defect in line with its expression at the distal ectoderm and pre-cartilaginous blastemal of the limb bud. Both fore and hind limbs were affected, displaying abnormal proximodistal patterning of all skeletal elements up to the level of the ilium/scapula which were normal. Precartilage condensation staining minimal separation of the three proximodistal segments from each other. The stylopod (humerus/femur) and zeugopod (ulna/radius or tibia/fibia) had a single primordium plus a poorly formed autopod (digits) region. This led to fusion of the stylopod and zeugopod with a thicker proximal region and no joint formation. Only 2–3 digits were formed with poorly distinguished carpal bones. A similar phenotype was seen in embryos treated with R115866. Increased RA availability in the absence of *Cyp26b1* led to loss of proximodistal patterning by the three distal Hox genes (*Hox12*, *a13* and *d13*) and expansion of *Meis 1* and *2* expression. Mesenchyme labelling experiments showed this led to presumptive autopod distal mesenchyme acquiring a more proximal identity thus contributing more to proximal (thickened) regions of the limb at the expense of the autopod in mutant limbs. The limb truncation phenotype was explained by greatly increased cell death and reduced chondrocyte maturation in the stylopod and zeugopod. Gene expression in the apical ectodermal ridge and zone of polarising activity was normal. [243,245]. 

Analysis of *Shh* loss-of-function mouse limb buds showed increased distal expansion of proximally located RA-dependent target genes alongside reduced distal *Cyp26b1* expression. Bead grafting experiments with FGF and FGF-inhibitor and AER-conditional mutant *Fgf* limbs showed *Cyp26b1* expression was dependent on FGF signals. This indicates that SHH-dependent AER-FGF signals likely control *Cyp26b1* expression to create an RA-free area in the distal limb. Mathematical modelling confirmed by RA-bead grafting experiments suggested an antagonistic AER-FGF-CYP26B1-RA signal linked to the SHH/GREM/AER/FGF feedback loop which enables SHH to promote distal limb development via CYP26B1-mediated RA degradation [244]. Combined with studies from the chick which implicate RA as a proximalising signal and AER-FGF as an antagonistic distalising signal upon limb mesenchyme fates, this led to the proposition of the “two-signal model” in which these signals control proximodistal limb development, together with WNT3 [204,334,335]. Further development of this model to better fit experimental results on PD patterning has led to addition of a cell-autonomous intrinsic timing element. *Hoxa13* expression is timed in limb mesenchyme in addition to cell cycle and duration of exposure to the AER signals to specify the stylopod and zeugopod/autopod. *Hoxa*13 expression is controlled by *Meis* genes via CYP26B1 degradation of RA and also requires limb cells to become competent for *Hoxa13* expression. Competence cannot be activated prematurely by experimental signalling methods. However, increased histone acetylation by inhibition of HDACs, making chromatin more accessible, did allow early expression of *Hoxa13*. This model integrates epigenetic regulation determining timed *Hoxa13* activation with classical limb patterning signals of SHH, FGF and RA and elucidates an epigenetically-regulated delay in gene expression timing required for spatial patterning of the limb [204,335,336]. Interestingly, the limb chondrocyte defects seen in *Cyp26b1^−/−^* mutants are compatible with this model. Deletion of *RARγ* in a *Cyp26b1* null background rescued stylopod and some digit formation, but not the zeugopod. This *RARγ*-mediated rescue of the apoptosis defect and chondrocyte differentiation improved limb mesenchyme survival but not proximodistal gene patterning disruption, uncoupling the chondrocyte defects from the patterning abnormalities [243] (Figure 6).

In addition to the limb defects other skeletal anomalies were observed in *Cyp26b1* mutant embryos, correlating with complex expression in cranial, pharyngeal and axial skeletal tissues including precartilage condensations, perichondrial cells and osteoblasts. Generally, expression in more mature ossifying cells was weaker than in less mature surrounding cells. Defective phenotypes included loss/underossification of both endochondral and inter-membranous bones including defects of the maxilla and mandible, calvarial bones, hyoid, thyroid and Meckel’s cartilage, fusion of the exocciptial and basioccipital bones, coronal synostosis and loss of the clavicle. Many of these structures are neural crest derived and reported caudal neural crest defects could contribute to this phenotype. However, mesodermally derived bones are also abnormal, suggesting that chondrocyte defect contributes to these phenotypes [202,246]. Further investigation in mice revealed genetic loss or chemical inhibition of *Cyp26b1* cells in limb mesenchymal cells resulted in reduced chondroblast differentiation, with cells instead being maintained in a pre-chondrogenic state accompanied by small increases in chondrocyte hypertrophy. There also appeared to be a small switch in fate towards a tendon lineage. Thus, *Cyp26b1* restricts chondrogenesis by limiting RA which promotes chondrocyte maturation and hypertrophy. Furthermore, R115866 treatment resulted in axial hyperossification and fusion of cervical vertebrae implying a role for *Cyp26b1* in spatial and temporal ossification patterning [202,242,337]. 

Three ENU *cyp26b1* loss-of-function mutations have been identified in zebrafish (*dolphin* and *stocksteif*) encompassing one splice-donor and two nonsense mutations. Phenotypes can be copied by morpholino injection and were similar to mouse and human mutant phenotypes. These defects included fusion/loss of the neurocranium and posterior pharyngeal arches. Hyperossification, premature osteoblast to osteocyte differentiation and excess mineralisation produced these phenotypes which could be phenocopied with exogenous RA and rescued by inhibition of RA signalling. This work revealed a role for *cyp26b1* in early neurocranial and pharyngeal patterning possibly via a mediolateral RA gradient and in inhibiting osteoblast maturation/activity and ossification [202,203,337]. 

The phenotypes from the various *Cyp26b1* mutants/chemical treatments described above were complex, varying with level of *Cyp26b1* depletion/RA augmentation and in some cases appearing to have opposing effects upon the same cells. For example, in zebrafish, coronal craniosynostosis resulting from ectopic suture mineralisation was observed, whereas in humans smaller, very fragile calvarial bones were seen. Further studies using RA/CYP26 inhibitor treatments in zebrafish revealed the same RA-induced premature osteoblast to preosteocyte transition at different time points in suture formation gave rise to both phenotypes. Premature osteoblast to osteocyte transition before suture formation caused reduced bone matrix production thus giving rise to smaller thinner calvaria. Preosteoctyes act to increase osteoclast activity via RANKL/Spp signalling and thus increase mineralisation/hyperossification. Regions of calvarial fragmentation displayed ectopic osteoclast activation and fragmentation could not be induced where there were no osteoclasts, suggesting bone resorption involvement. These phenotypes were associated with complete loss of *Cyp26b1* function acting before suture formation begins. In hypomorphic models, the effect of the premature osteoblast to osteocyte transition was felt after suture formation initiation, where increased mineralisation and thus premature suture mineralisation leading to suture was the result of decreased osteoblast offsetting of bone mineralisation by Ca^2+^ sequestration and increased mineralisation by osteocytes. Thus *CYP26B1/cyp26b1* mutants can therefore display phenotypes consistent with both net gain and net loss of bone [338] (Figure 6).

A role for RA/CYP26B1 in notochord epithelial chordoblast cells was shown to underlie axial hyper-mineralisation and vertebral fusion phenotypes with a similar mechanism to osteoblasts, including differentiation from bone-matrix producing cells to bone mineralising cells. *Cyp26b1* metameric expression in a subset of chordoblasts along the axis is required to prevent RA signalling in intervertebral spaces, restricting the establishment of biomineralising cells. If this RA repression is abrogated by loss of *cyp26b1* in this chordoblast subset, they ectopically acquire characteristics similar to preosteocytes, reducing matrix production and increasing mineralisation. This CYP26B1-mediated regulation of RA in chordoblasts seems required for the generation of a segmented vertebral column [339]. 

Finally, null mutations for *Cyp26a1* in the mouse also revealed a skeletal phenotype of variable expressivity. Skeletal abnormalities were frequent in *Cyp26a1^−/−^* mice. Posterior vertebrae (lumbar, sacral and caudal) were frequently missing, thoracic vertebrae were often deformed with fused ribs and cervical vertebrae (C1-7) exhibited variable posterior homeotic transformations. A small number of embryos (20%) exhibited sirenomelia due to hindlimb fusion, whilst some others exhibited malpositioning of the hindlimbs, but otherwise limb development was normal. To date, investigation into the underlying causes of these defects has been much less thorough than for *Cyp26b1* mutation skeletal phenotypes. However, it has been suggested that shifts in anterior and posterior RA gradients could underlie the *Cyp26a1* skeletal defects [179,231].

## 12. Pharyngeal and Cardiovascular Systems

Normal development of the pharyngeal system and the neural crest are important for cardiovascular development (reviewed by [340,341,342,343]). Retinoic acid signalling is necessary for this development to take place [12,13,14]. All three *Cyp26* genes are expressed in differing pharyngeal tissues during mouse development. At E9.5 and 10.5, there is strong expression of *Cyp26a1* in cervical and pharyngeal arch mesenchyme, including neural crest derived mesenchyme and the maxillo-mandibular cleft, whereas *Cyp26b1* is expressed in the pharyngeal surface ectoderm and pharyngeal pouch mesoderm and both genes are likely expressed in the developing SHF/OFT. *Cyp26c1* is seen in maxillary and mandibular tissues of pa1 including surface ectoderm, superficial arch mesenchyme and maxillo-mandibular cleft. Strong expression is also observed in epibranchial placodes as well as a domain encompassing the cervical mesenchyme of the head lateral and caudal to the otic vesicle and the caudal pharyngeal endoderm. 

All three genes are down/dysregulated in these tissues in *Tbx1* mutants which display loss of caudal pharyngeal arches/arteries/pouches and SHF/OFT defects, leading to cardiac defects affecting the great vessels and OFT. These defects are phenocopied in R115866-treated chick embryos and include interrupted aortic arch B, common arterial trunk, double outlet right ventricle and ventricular septal defects [126].

*Cyp26a1/c1* double null mutants have been reported to display neural crest defects. Premigratory neural crests were deemed to form normally in the neural plate but failed to undergo epithelial–mesenchymal transition correctly, particularly in more cranial regions, leading to reduced head mesenchyme, abnormal cranial ganglia and reduced size of pharyngeal arches [226]. 

*Cyp26b1* mutants have abnormal skeletogenesis, affecting head and neck as discussed above. These mutants have also been described with normal hindbrain patterning but disorganised neural crest streams affecting pharyngeal arches 3–6 and mild cranial nerve defects, although in mice *Cyp26b1* is not highly expressed in migrating neural crest, suggesting a possible non-cell autonomous effect from increased RA resulting from loss of *Cyp26b1* in neighbouring tissues such as the pharyngeal endoderm/ectoderm [246]. Knockdown of *cyp26b1* in the zebrafish also affects cranial nerves, particularly the vagus, in addition to jaw and gill neural-crest derived cartilages [248]. Expression of zebrafish *cyp26b1* is also required in non-tendon neural crest cells separating two *scxr*-positive tenoblast populations to allow tenoblast to tendon condensation and thus normal musculoskeletal patterning of pharyngeal arch 1 and 2 tendon and muscle attachments. In the absence of *cyp26b1* movement of tenoblasts into mature tendons is disrupted leading to ectopic muscle projection, which finally associate with areas of ectopic tendon marker expression. Thus, a non-cell autonomous role for *cyp26b1* in neural crest regulates pharyngeal tendon development [344]. *Cyp26b1* is also important in tongue muscle and palate formation in the mouse. In *Cyp26b1^−/−^* embryos, palatal shelves fail to elevate and have reduced expression of genes associated with proliferation (*Fgf10*, *Tbx1*, *Bmp2*) and reduced proliferation in the region of the palatal bend. Tongue muscles were a/hypoplastic likely affecting tongue withdrawal necessary for palatal closure. Ectopic RA signalling was observed in neural-crest derived mesenchyme surrounding affected tongue muscles, which form part of the pharyngeal arch 1 and 2 derived musculature [247]. SHH signalling is also implicated in palatal development, with *Cyp26a1* and *Cyp26b1* expression decreased and a concomitant increase in RA signals in *Shh* LOF mutants displaying cleft palate phenotypes [298]. The same mechanism in the tongue epithelium of *Shh* LOF mutants led to change in cell fate specification causing the formation of ectopic salivary glands and increased size taste-buds. Thus, *Shh*-mediated activation of *Cyp26a1/c1* acts to control RA availability in these tissues [345]. 

Regulation of RA in the pharyngeal endoderm is necessary for thymus development [68,107]. *Cyp26b1* mutants present with ectoptic and small/missing thymus at mid-gestation, a phenotype also seen in *Tbx1* mutant mice [118,246,346] (C. Roberts unpublished data). Furthermore, *Cyp26b1* RA-regulation seems to be required for regulation of RA-driven signalling in activated T-cells in gut-related lymphoid tissues [195].

As described above, loss-of-function phenotypes for the CYP26 enzymes individually and together can result in phenotypes affecting pharyngeal development, the neural crest and the cardiovascular system. The initial papers describing deletion phenotypes do not describe the defects affecting these developmental systems in great detail. For *Cyp26a1* in the mouse this comprises only the information that some embryos were developmentally arrested between E8.5 and 9.5 and showed cardiac looping and dilation defects [179,231]. 

However, further studies in the zebrafish have revealed important roles in cardiac lineage commitments within the early anterior lateral plate mesoderm (APLM) and requirements for ventricular cardiomyocyte contributions from both first and second heart field sources [251,347]. 

Knock-down of *cyp26a1* or *cyp26c1* individually does not reveal a significant cardiac defect but depletion of both produces a number of anomalies affecting cardiac development and upregulated expression of RARE-reporters indicating increased RA levels. Both genes are expressed in the APLM at 8 somites with a significant overlap of expression with vascular progenitor (VP) marker *etv2*, but a minimal overlap of the posterior expression boundary with the anterior limit of *nkx2.5* marking cardiac progenitor (CP) cells. In double-deficient *cyp26a1/c1^−/−^* embryos at 36hpf atrial cardiomyocyte number was increased in manner reminiscent to that seen with moderate RA treatment [251,348,349,350]. At 8 somites, ventricular and markers of atrial and ventricular progenitor markers (*nkx2.5/hand2*) (no atrial specific markers are yet identified for this stage) were shifted anteriorly in a fashion suggesting a possible expansion of atrial CPs. 

RA-responsive *dhrs3a* is expressed posterior to CP markers in the APLM [44] and in *cyp26a1/c1* LOF embryos is expanded with a forward shift of the anterior boundary whilst the overall size of the APLM was unchanged. These changes in the patterning of the APLM could be restored by moderate inhibition of RA signalling. Lineage tracing revealed that in wild types anterior APLM gave rise to endothelial/endocardial cells whereas more posterior APLM produced myocardial cells. In *cyp26a1/c1*-deficient embryos labelled cells fated to become endothelial cells were significantly less frequent and endothelial/endocardial progenitors were not found posteriorly. In contrast, the origins of chamber myocardial progenitors expanded to include much more anterior APLM and the frequency of labelled atrial progenitors increased whereas that of ventricular progenitors was similar to controls. Cranial endothelial and endocardial cell contribution was greatly reduced and displayed abnormal morphology. Together, these data suggested expansion of atrial specification at the expense of the vascular lineage. Finally, transplantation experiments revealed that *cyp26a1/c1* regulation of RA-mediated APLM patterning is non-cell autonomous [251]. 

Analysis of these *cyp26a1/c1*-deficient embryos and CYP26-inhibitor (ketacozanole) treated embryos at later stages (48-2hpf) identified reduced numbers of ventricular cardiomyocytes and markers of cardiac differentiation, despite previous data suggesting that specification of ventricular cardiomyocytes was relatively unchanged. These ventricular defects were independent of the increased atrial specification anomalies described above and could be produced with lower MO concentrations than required for the early APLM patterning defects, therefore suggesting that this abnormality arose after formation of the linear first heart field-derived heart tube. Instead, photo-conversion studies showed the addition of second heart field (SHF) cells to the arterial pole where they contribute to the ventricle and outflow tract was disrupted. SHF patterning was normal in double-deficient embryos but undifferentiated SHF-progenitors accumulated abnormally outside the heart tube. This aberrant migration then led to failure to contribute to OFT and ventricle, with SHF cells instead participating in pharyngeal arch artery 3/4 endothelium formation. Furthermore, reduced proliferation and expression of FGF8 was observed in *cyp26a1/c1* knockdown embryos. This SHF-driven part of the ventricular loss could be partially rescued by restoring expression of FGF8 which is required for SHF progenitor proliferation [347].

However, this SHF-defect alone was not sufficient to account for the total loss of ventricular cells that were observed, nor were changes in apoptosis seen. Instead, ectopic cardiomyocytes, likely arising from the FHF, were seen outside the heart. Time-lapse imaging showed that these cells were extruded from the differentiated heart tube in around half of the *cyp26a1/c1*-deficient embryos. These ectopic clusters then underwent apoptosis, thus contributing to the overall loss of ventricular cardiomyocytes. This phenomenon may be regulated by aberrant polarity and adhesion with *cyp26a1/c1*-LOF ventricular cells appearing rounder with mis-expressed ZO1 and Β-CATENIN. This extrusion defect was not repaired by restoring FGF8 expression. Interestingly, MMP9 was found to be increased in *cyp26a1/c1*-deficient hearts. Treatment of wild-type embryos induced a small number to display ventricular phenotypes similar to loss of *cyp26a1/c1*. Moreover, when *cyp26a1/c1*-deficient embryos were treated with an MMP inhibitor, both SHF and cardiomyocyte extrusion phenotypes were found to be reduced without any amelioration of loss of FGF8 expression, suggesting that these molecules act in parallel in this context [347].

Recent studies have shown RA can specify atrial cardiomyocyte fate in human PSCs [351,352]. Embryoid bodies generated from hPSC cells treated with RA can generate both atrial and ventricular cardiomyocyte fates from different mesodermal precursors. Ventricular cells seemed to arise from CD235-positive mesodermal precursors which express high levels of *CYP26A1* on day 3 of differentiation, with low levels of *RALDH2* expression. Atrial cardiomyocytes arose from mesoderm expressing high levels of *RALDH2* but no *CYP26A1* from day 2–3 of differentiation. These cells responded to retinol via autocrine signals to produce atrial cardiomyocytes, apparently at the expense of ventricular fate. In summary, inhibition of RA signalling by *CYP26A1* is required for CD235+ mesodermal precursors to give rise to ventricular fates whereas atrial induction is mediated via autocrine RA signalling in *RALDH2*+ mesoderm. Thus, the balance between atrial versus ventricular fate induction is, in part, mediated by balance between RA synthesis (*RALDH2*+) and degradation expression (*CYP26A1*+) in two mesodermal precursor populations in a similar fashion to that exhibited in embryos [251,353], reviewed [354]. 

Cardiac development was reported as normal in *Cyp26b1* mutants by Maclean et al. 2009 [246]. However, this report may be the result of mouse strain genetic background/deletion allele used, as unpublished data using the *Cyp26b1* deletion allele published by Yashiro et al. 2004 [245], show a variety of cardiac defects, affecting the arterial pole, atrioventricular septation and epicardial/coronary vessel development (C. Roberts unpublished data, K. Yashiro pers.comm).

## 13. Other Systems

Foregut endoderm caudal to the pharyngeal endoderm is patterned by RA generated in the adjacent dorsal mesoderm to give rise to normal pancreas development. In zebrafish, expression of *cyp26a1* is necessary to define the normal anterior limit of the pancreatic field by a feedback loop regulating expression of RA within the endoderm. The posterior limit is set by *Cdx4* expression. Loss of *cyp26a1* leads to anterior expansion of the pancreatic anlagen at the expense of more anterior (pharyngeal) endoderm derivatives [181,355]. In the developing zebrafish kidney, a similar mechanism specifies the dorsoventral axis. RA on the dorsal side of the embryo specifies anterior kidney fates, whereas expression of *cyp26a1* ventrally is necessary to inhibit RA signalling allowing the induction of ventral kidney progenitors [356]. Moreover, the *cdx1a* and *cdx4* regulate the posterior boundaries of expression of *raldh2* and *cyp26a1* in the anterior paraxial mesoderm to correctly position the zebrafish pronephros along the anterior–posterior axis [357].

Patterning within the splanchnic mesoderm by *Pbx1-Tlx1* (*Hox11*) gives rise to the splenic progenitor cells [358]. *Tlx1* regulates proliferation of the splenic mesenchyme. It has both repressor and activator functions and in *Tlx1* mutants RA signalling pathway genes such as *Raldh1*, RA nuclear receptors and Vitamin A/RA transporters are up-regulated, whilst RA-degrading *Cyp26b1* was down-regulated. In *Cyp26b1* mutants, the splenic primordium mesenchyme is greatly reduced in size, suggesting that regulation of RA availability by *Cyp26b1* is essential for spleen precursor cell proliferation and maintenance, particularly as in *Tlx1* mutants, excess RA led to premature differentiation of this population [359]. 

Oedema and haemorrhage characterise the external phenotype of the *Cyp26b1* mutant mouse. Whilst cardiac defects may contribute to these phenotypes, a severe lymphatic defect is also clearly causative. Expansion of lymphatic endothelial cell (LEC) progenitors arising from cardinal veins is seen in E11.5 mutants likely as a result of the loss of *Cyp26b1* in *Prox1*-positive LECs on the dorsal side of the cardinal vein. For both deep and superficial lymphatics (dermal and jugular), increased size of lymph sacs and lymphatic vasculature was seen with reduced branching. Subcutaneous oedema presented at later stages along with accumulation of blood in lymphatic vessels. This was ascribed to the expanded size and increased expression of FOXC2 within the lymphovenous valves which normally separate vascular and lymphatic networks. Gain-of-function *Cyp26b1* embryos exhibited reverse phenotypes with hypoplastic development of the LEC progenitor pool and lymph sacs [360]. Thus, *Cyp26b1* regulation of RA availability controls the number of LEC progenitors during development.

*Cyp26b1* regulation of RA levels is also required for normal skin and hair development. Expression begins in the mesenchyme around hair follicles at E14.5 and is strongly expressed in the dermis but not the epidermis by the end of development. Loss of this expression leads to reduction of the cornified envelope layers (terminally differentiated keratinocytes) and arrest of hair follicle development at the germ stage and led to increased proliferation in the basal layer. Keratin19 is abnormally upregulated from zero in the epidermis in *Cyp26b1* mutant embryos and RARβ in dermis and epidermis. Genes associated with development of the cornified layers and known to be upregulated in cases of skin barrier disruption were upregulated and increased skin barrier permeability was demonstrated. Keratins associated with hair follicle development were down-regulated. Importantly, *filaggrin* expression was completely lost. However, conditional deletion of *Cyp26b1* in the dermal layer alone, did not recapitulate the epidermal phenotype. Markers of the peridermis, which is normally removed via desquamation were strongly upregulated and shown to be both inducible by RA and to be linked to *filaggrin* downregulation, suggesting that the failure to form a proper epidermal barrier was linked to abnormal peridermal development. Effects on hair follicle development included decreased hair follicle density and markers of follicle differentiation were down-regulated and differential expression of factors linked to hair follicle development including the *Wnt* pathway, and *Runx*, *Sox* and *Fox* families was observed [361]. 

Retinoic acid is also essential for the process of haematopoiesis, where haematopietic stem cells (HSCs) are specified in the aorta-gonad-mesonephros, leading to the birth of HSCs from hemogenic/aortic endothelium, with WNT signalling blocking the inductive effects of RA [362]. So far, no evidence has been provided as to a developmental role for CYP26s in this process. However, in vitro, away from the stem cell niche, bone marrow-derived *Raldh1*-positive primitive haematopoietic cells have been shown to terminally differentiate rapidly, being unable to maintain their self-renewal. Inhibition of RA signalling restored the initial phenotype and enable self-renewal. It was shown that bone marrow stromal cell expression of *Cyp26a1/Cyp26b1* mediated this RA inhibition driven phenotype. Therefore, prevention of intrinsic RA-driven HSC differentiation requires degradation by bone marrow CYP26 enzymes [363]. 

Extensive experiments suggest that RA from the developing mesonephros region is required to induce meiosis in adjacent embryonic germ cells. However, significant differences in the timing of meiosis onset are apparent between male and female gonads. In female mice, meiosis is initiated at E13.5 (mid-late gestation), whereas males do not enter meiosis until post-natal stages. Pan-RAR antagonists prevent expression of RA-dependent *Stra6* expression in cultured ovaries, whereas exogenous RA could induce expression in cultured embryonic testes which normally remain *Stra6*-negative. Thus in ovaries RA can induce *Stra6* expression but testes are protected from this RA-activity. *Cyp26b1* is expressed in the developing male gonad but not in embryonic ovaries and was discovered to be the RA-protective agent in the developing testis. When CYP26 expression is inhibited by chemical blocked (ketacozanole and R115866), cultured testes express *Stra6* within germ cells. This up-regulation is prevented by inclusion of RAR-antagonists, confirming that *Stra6* expression is RAR-signal dependent. *Cyp26b1* null mice displayed premature entry into meiosis, similar to that observed in normal female gonads [364,365,366].

Normal male germ cells do not begin meiosis, instead becoming quiescent until puberty by supressing mitotic proliferation. RA is known to act as a driver of both mitosis and meiosis, thus testicular germ cells need to be protected against both these functions of RA until the appropriate developmental time-point. *Cyp26b1* is present in the somatic cells of the testis degrading paracrine RA signals from near-by tissues such as the mesonephros. Studies in double knock-out *Cyp26b1/Stra6* mice revealed that this RA-metabolising function of testicular *Cyp26b1* acts to prevent the initiation of *Stra6*-dependent meiosis by RA and also prevents the *Stra6*-independent activation of mitosis. This suppression of mitosis appears to require cyclin-dependent kinase inhibitors to arrest germ cells in the G0/G1 phase. Therefore, *Cyp26b1* is required to regulate both RA-driven testicular meiosis and mitosis in male germ cells [366,367,368]. 

## 14. Regeneration

In addition to the roles regulating developmental progenitors, iPSC and HSC cells, *Cyp26a1* and *Cyp26b1* have been shown to play multiple roles during regeneration of the adult fin in zebrafish and lens in *Xenopus*. In the former, the *cyp26* genes are required to regulate both non and cell-autonomous signalling. Following fin amputation, regeneration requires the formation of a blastema composed of cycling progenitor cells at the injury site. The formation, proliferation and maintenance of the blastemal requires a huge increase in RA signalling from fibroblast cells which is induced by amputation. To replace the lost bone elements of the amputated fin mature non-proliferative osteoblasts transition by dedifferentiation to a preosteoblast profiferating state to form a progenitor pool within the blastema. Following several cycles of division, these preosteoblasts redifferentiate to produce new bone. However, this de and re-differentiation is inhibited by RA, thus the high RA levels necessary for the blastemal conflict with osteoblast regenerative mechanisms [369,370]. 

Analysis of *cyp26b1* and *raldh2* expression in experimental situations where RA levels in both the uninjured and amputated fin were experimentally manipulated revealed that *cyp26b1* is initially upregulated in stump osteoblasts during limb regeneration. This then allows osteoblast dedifferentiation. *Cyp26b1* levels are then down-regulated to promote RA-driven proliferation of the newly formed preostoblasts. After several rounds of proliferation, the preosteoblasts redifferentiate in a proximal-to-distal direction to give rise to non-proliferative osteoblasts producing new bone. Within the blastemal fibroblasts, *cyp26b1* and *raldh2* exhibit opposite gradients of expression: R*aldh2* is high distally and rapidly decreases proximally while *Cyp26b1* is absent in distal fibroblasts but expressed widely proximally. Proximal *cyp26b1*-positive cells may therefore act as an RA sink to generate RA gradients. High RA distally promotes preosteoblast proliferation and inhibits redifferentiation in this region. Varying the proximodistal position of amputation changes the RA gradient produced and gave rise to distal limits of preosteoblast redifferentiation consistent with the idea that below a certain RA threshold proximally preosteoblasts cease proliferating and redifferentiatiate to mature bone-producing osteoblasts [371]. 

Expression of *cyp26b1* in both stump osteoblasts and fibroblasts is key in protecting the osteoblasts from high RA levels generated by blastemal fibroblasts to allow bone regeneration. It has been shown that WNT/Β-CATENIN signalling promotes proliferation towards the distal tip of the regenerating fin, whereas BMP expression in differentiating osteoblasts stimulates differentiation, partly by inhibiting WNT signalling. Markers for BMP and WNT-inhibitors are down-regulated shortly after exposure to RA treatment. This implies that RA may have a negative impact on BMP signalling and promote WNT signalling, thus inhibiting preosteoblast redifferentiation [371].

A similar precise regulation of RA levels along the regenerating fin may also be required for osteoclast function. Osteoclasts are required for bone resorption in normal fin development and are not present in the uninjured adult fin. However, they are present in the regenerating fin. RA has been shown to inhibit osteoclast differentiation and excess bone matrix has been observed in regenerates exposed to excess RA promoting the idea that excess RA inhibits osteoclast differentiation and thus bone resorption [371].

In summary, *cyp26b1*-mediated degradation of RA acts to regulate both non- and cell-autonomous RA signalling during fin regeneration. Opposing *raldh2* and *cyp26b1* expression within fibroblasts establishes an RA environmental gradient with high RA distally and low RA proximally. Transient cell-autonomous degradation in osteoblasts by *cyp26b1* allows dedifferentiation to occur. Inactivation of RA controls the preosteoblast proliferation rate and the PD axis position where preosteoblasts redifferentiate into mature osteoblasts. In conclusion, regulation of RA signalling by *cyp26b1* controls bone matrix synthesis by osteoblasts and resorption by osteoclasts and thus drives the regeneration of the amputated fin [371] (Figure 7).

In addition to controlling regenerative outgrowth of bone CYP26-mediated protection from RA signalling is also necessary to establish ray-interray organisation in the regenerating fin. In this scenario, *cyp26a1* degradation of RA in the proximal basal epidermal layer is crucial for spatial restriction of osteoblasts. *Cyp26a1* and *shha* are expressed within the basal epidermal layer next to pre and differentiating osteoblasts. With both exposure to exogenous RA and experimentally reduced CYP26 function *shha* and its receptor *ptch2*, are down-regulated, indicating that expression of these genes requires a low RA niche. In these regenerates, preosteoblasts migrate into the interray regions that they are normally excluded from, indicating a failure to align at the proximolateral blastema and disruption of ray-interray boundaries. Similar preosteoblast ectopic location in the interray leading to ray fusion can be produced by inducing *cyp26a1* domains in abnormal proximity to each other, thus altering the patterning of RA-responsive/non-responsive regions within the fin. 

Other blastemal cell types also spread ectopically into the interray regions and blood vessels also ignore the usual restrictions of regenerative patterning to form connections across interrays with adjacent rays. This leads to extensive disruption of ray-interray patterning including ectopic bone formation and a permanent repression of fin regeneration. Experiments altering exposure of pre/osteoblasts to *shha* signalling suggest that *shha* signals from *cyp26a1*-positive cells within the basal epidermis promote adjacent osteoblasts to proliferate. Furthermore FGF-dependent exclusion of *shha* from distal regions was found to be dependent upon inhibition of *cyp26a1* expression. Therefore, *cyp26a1* expression within specific domains of the basal epidermis creates regions protected from RA signalling thus allowing signalling required for the proper alignment of preosteoblasts in the developing blastema, which is important for correctly establishing ray-interray boundaries within the blastemal. Furthermore, *cyp26a1*-mediated RA degradation within the basal epidermis is necessary to allow *shha*-driven proliferation in neighbouring preosteoblasts during regeneration [372,373].

A role for CYP26 enzymes in the regeneration of the lens in *Xenopus* larvae and hair cells has also been documented. Following the loss of a lens, the epithelial cornea can become competent to regenerate the lens with FGF signalling from the neural retina playing a key role. In control and regenerating, corneal epithelium members of the *Raldh* and *Cyp26* gene families are expressed. Interestingly, inhibition of CYP26 enzyme function, exogenous RA treatment and a synthetic retinoid not processed by CYP26 enzymes all repress lens regeneration, with CYP26 antagonism leading to reduced corneal proliferation and failure to form a lens. In addition, expression of *cyp26* genes may be important to maintain expression of key lens genes such as *pax6* and *fgfr2*. The requirement for *cyp26* expression in lens regeneration is from 12 to 48 h post ablation, suggesting a possible role in establishing corneal competency to respond to lens inducing signals [374,375]. 

In zebrafish, inner ear *raldh3* is expressed in the anterior of the posterior macula sensory patches with *cyp26b1* expressed in a complementary fashion, posteriorly. In the lateral crista, a similar complementary expression is seen, this time across a mediolateral axis. These complementary expression domains likely generate RA gradients across these tissues, with *rarα* expression seen equally across the region. Following laser ablation of hair cells the regenerative response in supporting cells includes upregulation of all genes in tissues already exhibiting expression and expansion of *raldh3* expression into previously negative areas. In the regenerating lateral line upregulation of *raldh3*, *rarβ* and *cyp26a1* followed by *cyp26c1*, is rapidly and transiently induced. *Cyp26a1* expression restricted to two cells adjacent to the neuromasts and *cyp26c1* is mostly seen in the dorsoventral poles of the neuromast. Suppression of *p27^kip^* and *sox2* expression is necessary following ablation to allow support cells to re-enter the cell cycle followed by subsequent differentiation to generate new hair and supporting cells. RA was shown to mediate this down-regulation and *cyp26* expression is limited to specific cells. However, the precise role of *cyp26* expression within specific cell populations during hair cell regeneration remains to be elucidated [376]. 

## 15. Paradoxical Feedback Loops in RA Signalling

Recent work in mouse and zebrafish has suggested the existence of feedback loops regulating retinoic acid signalling which can lead to apparently paradoxical phenotypes, whereby excess RA/RA signalling will produce both gain and loss of function phenotypes and vice versa. 

Using the mouse kidney as a model, Lee et al. 2012 [377] showed that a teratogenic dose of RA at E9.0 led to a transient increase and ectopic location of the expression of RA-degrading CYP26A1 and CYP26B1 levels, concurrently with a longer-lasting reduction in the RA synthesising RALDH1-3 enzymes. HPLC quantification showed an initial rapid up-regulation of RA shortly after the RA insult followed by a longer-term overall reduction in RA levels. Supplementation with low doses of RA in the time period showing reduced RA following the initial high dose insult allowed rescue of the renal agenesis phenotype mediated by RA-driven reduction of *Wt1*. Development of other organs including the eye, ear, jaw, heart, and hindlimb were also improved by this subsequent low-dose RA rescue regimen although tail agenesis, palate defects and imperforate anus were not rescued. These results led to the conclusion that an excess RA insult leads to a feedback mechanism to reduce RA levels which eventually produces an RA deficiency. Thus, the defects observed are the result of early excess RA signalling in affected tissues compounded by prolonged RA deficiency induced by the mechanism to remove excess RA.

Interestingly, similar feedback mechanisms were observed in RAR-depleted and hyperactive RAR models in the zebrafish. Depletion of a novel conserved RAR–splice variant *rarαb1*, thereby reducing RA signalling, led to increased heart size via increased specification of cardiomyocytes. This defect was driven via a positive feedback mechanism induced by loss of *rarαb1,* which eventually gave rise to increased RA signalling via other RARs and up-regulation of RA-responsive genes including *hox5b*, very similar to those observed with exogenous RA treatment. This increase in embryonic RA was confirmed by increased expression in an RA sensor transgenic line and was driven by increased expression of *rdh10a* and *rdh10b*. *Cyp26a1* as an RA-responsive gene was also upregulated to protect the *rarαb1*-depeleted embryos from teratogenic RA-signalling. Therefore continuing reduction of *rarαb1* causes an over-active positive response from RA levels and RA signalling producing RA-driven teratogenic phenotypes [350,378]. 

The reverse experiment with gain-of-function of RA signalling was produced using a hyperactive RAR transgenic zebrafish line. Again, both gain and loss of function phenotypes were observed affecting anterior tissues such as the midbrain and more posterior organs including the heart. Similarly to the mouse exogenous RA experiment described above, gain-of-function phenotypes were due to the initial increase in RA signalling, with the loss-of-function defects arising from the subsequent fast induction of increased *cyp26a1* in response to the increased RA signalling which led to an overall reduction in embryonic RA as shown by an RA-responsive transgenic reporter line. Transplantation experiments of the RA sensor line showed that that *cyp26a1* can act as an RA sink with non-cell autonomous effects upon local RA levels and thus patterning gradients [378,379]. 

## 16. Conclusions

Retinoic acid signalling controls a variety of important processes during development and regeneration, including transcriptional patterning events, proliferation and differentiation necessary for the normal development of many organ systems. RA functions as a morphogen, with both non-cell and cell-autonomous roles, with complex regulatory feedback mechanisms. A growing understanding that regulating RA availability is key to regulating RA signalling has led to an increasing body of work in the last decade concerning the enzymes that synthesise and degrade RA. It is becoming increasingly apparent that RA synthesis and degradation are frequently coupled in opposing directions to pattern the developing embryo. This has produced a rising appreciation of the role of the CYP26 RA-degrading enzymes. As previously suggested, CYP26 enzymes do act as RA sinks and protect the developing embryo against excessive exposure to RA. However, their function extends well beyond this simple model. In multiple systems, complex dynamic expression of *Raldh* and *Cyp26* genes allows subtle and often apparently contrary RA-signalling in both a cell and non-cell autonomous fashion across different cell types within the same and adjacent tissues. This provides mechanisms for a variety of processes including protection of specific cell types against a background of high environmental RA, creating RA gradients necessary to establish transcriptional pattern across developing organs, regulating equilibrium in bipotential progenitor cell fate and establishing and maintaining sharp developmental boundaries. Much remains to be understood as to the function of these enzymes in regulating RA availability in both development and regeneration, and also in terms of human health, with regard to congenital disease, understanding possible roles in cancer causation and cure and in possible future regenerative therapies.

## Figures and Tables

**Figure 1 jdb-08-00006-f001:**
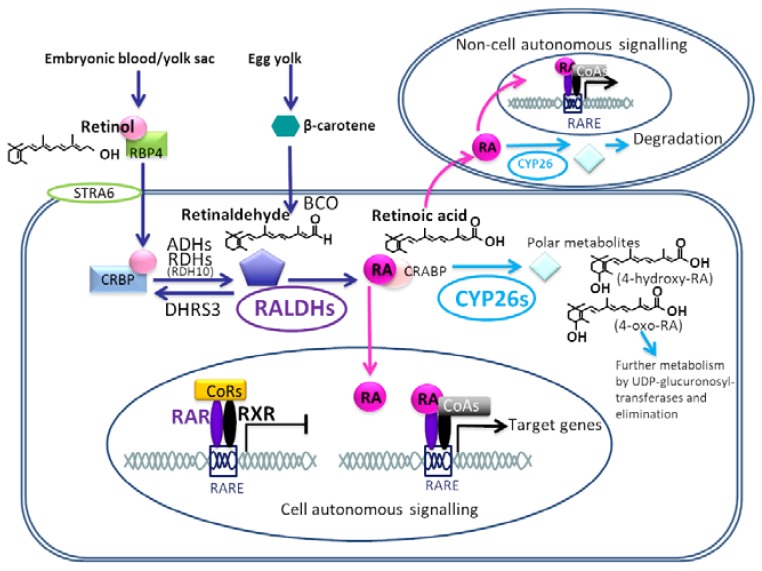
Schematic of the retinoic acid signalling pathway during development. Maternal diet-derived retinol/ in blood/yolk sac/yolk of embryo bound to retinol binding protein 4 (RBP4) enter cellular cytoplasm via binding to membrane bound RBP/RA complex receptor STRA6 (Stimulated by Retinoic Acid 6). Retinol is then bound to cellular retinol binding protein (CRBP) and reversibly oxidised to the intermediate form retinaldehyde (also known as retinal) by alcohol dehydrogenase and retinol dehydrogenase (particularly RDH10) enzymes. The reverse reaction, retinaldehyde to retinol is catalysed by the dehydrogenase/reductase 3 (DHRS3) enzyme. Retinaldehyde can also be generated from β-carotene by β-carotene 15,15′-monooxygenase (BCO). Retinaldehyde is then irreversibly converted to retinoic acid (RA) by retinaldehyde dehydrogenase (RALDH) enzymes, particularly RALDH2. RA can then undergo three different processes: (1.) RA bound to cellular retinoic acid binding proteins which shuttle RA to the nucleus. Hetero-dimerised RAR-RXR complexes (retinoic acid receptor-retinoid-X-receptor) are bound to conserved retinoic acid responsive elements (RARE) within the promotors of target genes. Most frequently, in the absence of RA co-repressor complexes (e.g. NCoR/SMRT) are bound to the RAR-RXRs, preventing transcription. Upon RA binding, the receptors undergo a conformational change, releasing co-repressor complexes and recruiting co-activator proteins (e.g. SWI/SNF, pCIP/p300, PolII) as replacements, thus triggering transcription activation of target genes. (2.) RA produced in one cells can also signal in a paracrine fashion to neighbouring cells, mediating non-cell autonomous effects. (3.) If Cytochrome P450 subfamily 26 (CYP26) enzymes are present in the cell, RA is hydroxylised in the cytoplasm to more polar metabolites with less biological activity, which are further processed by UDP-gluconyl transferases and eventually eliminated from the cell. Adapted from Niederreither and Dolle 2008 [10].

**Figure 2 jdb-08-00006-f002:**
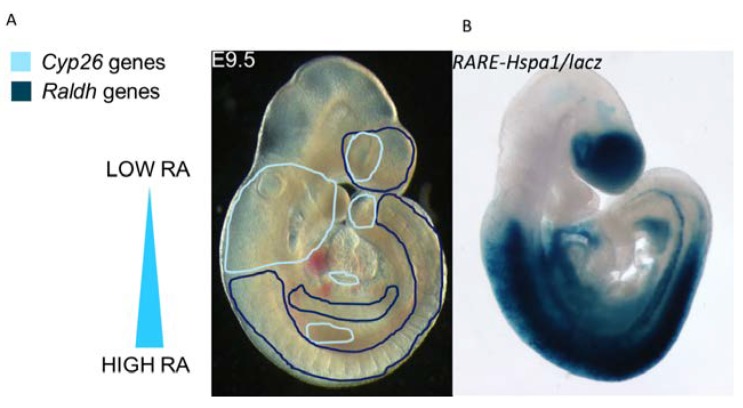
Overview of expression of retinoic acid regulating enzyme expression and retinoic acid signalling in mouse E9.5 embryos.(**A**). Combined expression domains of RALDH1-3 retinoic acid synthesising enzymes (dark blue lines) versus CYP26A1, B1 and C1 retinoic acid degrading enzymes (pale blue lines) depicted on an E9.5 embryo. RA synthesising activity is mostly localised in the caudal part of the embryo due to RALDH2 expression in the paraxial and splanchnic mesoderm. RALDH1 and 3 contribute to small domains of expression around the developing eye and forebrain. CYP26 expression is mostly localised to the pharyngeo-cardiovascular tissues and hindbrain, with small regions of expression in the tail-bud and eye. This establishes a high to low postero-anterior RA gradient across the embryo, with varying expression in specific tissues modulating the exposue to RA further. (**B**). Beta-galactosidase staining in a wild type E9.5 *RARE-Hspa1b/lacz* transgenic mouse, giving a read-out of regions of retinoic acid signalling at this stage of development. Overall, regions of active RA signalling correspond to regions expressing RALDH RA synthesising enzymes, whereas those tissues expressing RA-degrading CYP26 enzymes are negative for RA signalling.

**Figure 3 jdb-08-00006-f003:**
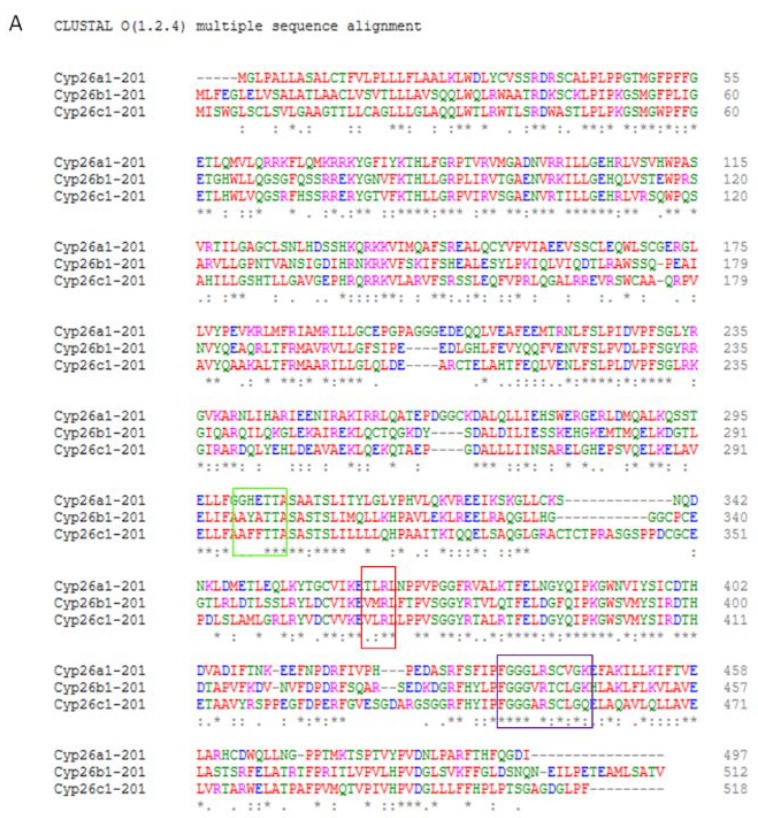
CYP26 protein amino acid sequences and protein schematic. (**A**). Clustal Omega alignment of mouse CYP26A1, B1 and C1 amino acid sequences showing the relatively low level of sequence conservation between the family members. (**B**). Clustal Omega alignment of amino acid sequence of CYP26B1 between zebrafish, mouse and human showing a high degree of conservation between species. (**C**). Schematic of CYP26 protein structure with conserved I (green box) and K (red box) helix regions and the conserved haem-binding domain (purple box). Same colour boxes denote these regions in (**A**,**B**).

**Figure 4 jdb-08-00006-f004:**
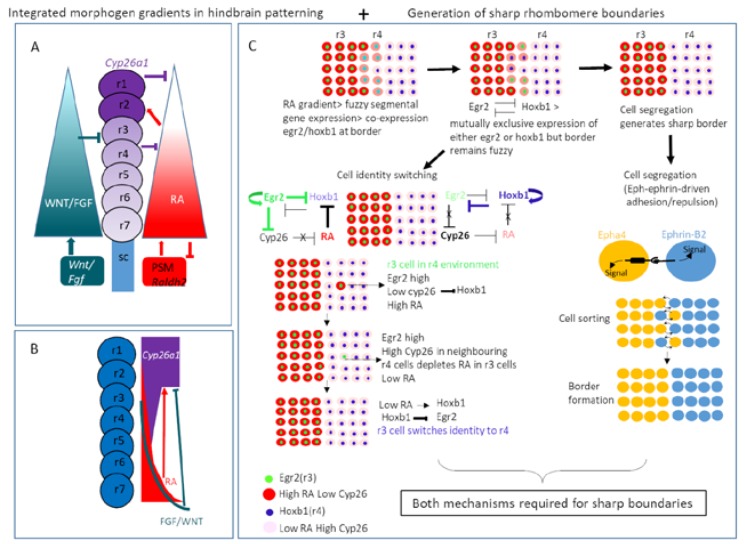
Regulation of RA availability by CYP26 enzymes in morphogen patterning of the hindbrain and generation of sharp rhombomere boundaries. (**A**) and (**B**). Integrated morphogen gradients in hindbrain patterning. Domains of RA synthesising enzyme *Raldh2* in the pre-somitic mesoderm at the posterior of the hindbrain and high RA-degrading enzyme *Cyp26a1* in the anterior neuroectoderm of the hindbrain plus lower *Cyp26a1/c1* in more posterior hindbrain combined with inhibitory signals from an FGF/WNT signal gradient establish an RA gradient which is robust to RA fluctuation and adapts as the hindbrain grows larger (adapted from White et al. 2008 [159]). (**C**). Generation of sharp rhombomere boundaries requires Eph-Ephrin-mediated cell segregation and a cell identity switching mechanism driven by the RA status of the local environment. Cells of high *egr2*, low *cyp26b1* and high RA specify r3 identity. If these cells are present in an r4 environment, high *cyp26b1* in surrounding r4 cells leads to depletion of RA in the r3 cell, this induces *hoxb1* expression which represses *egr2* leading to a switch to r4 identity of high *cyp26b1*, low RA, high *hoxb1*. *Hoxb1* and *egr2* reciprocally repress each-others’ expression maintaining identity in each rhombomeric segment (adapted from Addison et al. 2018 [293], Wilkinson 2018 [294] and Kitazawa and Rijli 2018 [292]).

**Figure 5 jdb-08-00006-f005:**
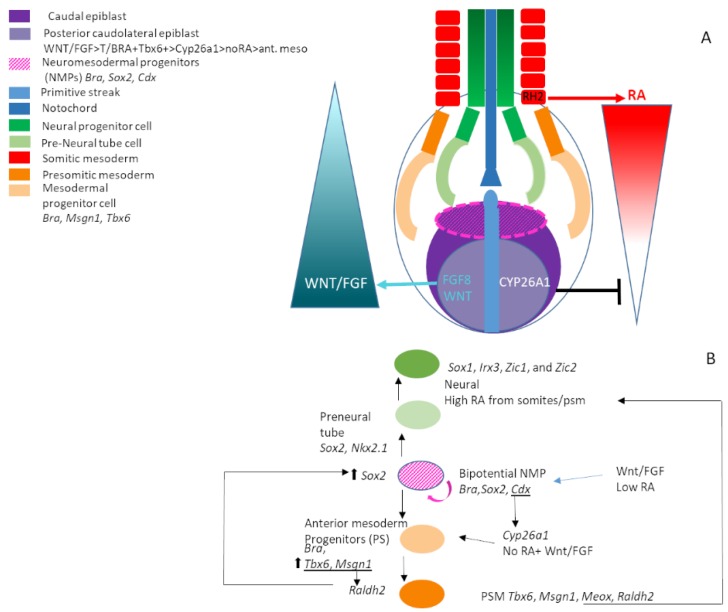
CYP26 enzyme function in neuromesodermal progenitor and axial elongation during development. (**A**) Schematic of posterior axial development, neuromesodermal progenitors and morphogen gradients. Expression of *Cyp26a1* in the posterior caudolateral epiblast in combination with FGF/WNT signalling and RA synthesis by *Raldh2* in the somatic mesoderm and epiblast establishes a posterior RA gradient required to promote caudal axial elongation. Axial elongation requires the establishment and maintenance of a neuromesodermal progenitor pool in the caudal epiblast at the anterior end of the regressing primitive streak. Bipotential NMPs can differentiate either to neural or mesodermal specifications and a balance between these fates is required for normal axial development. (**B**) Schematic of the molecular signals governing bipotential NMP fate specification towards either neural or mesodermal tissue. Posteriorly, CYP26A1-mediated inhibition of RA plus FGF/WNT signalling induces *Bra/Tbx6*-positive mesodermal tissue from NMPs. Anteriorly high RA promotes induction of *Sox2/Nkx2.1* pre-neural fates from NMPs. NMPs themselves (*Bra/Sox2*-positive) require low levels of RA for induction and maintenance. In this model, the relative levels of mutually antagonistic RA and WNT signalling control *T/Bra* and *Sox2* expression and regulate the switch between mesodermal and neural differentiation and therefore the proportion of the two cell fates. Adapted from Niederreither and Dolle 2008 [10] and Gouti et al. 2017 [330].

**Figure 6 jdb-08-00006-f006:**
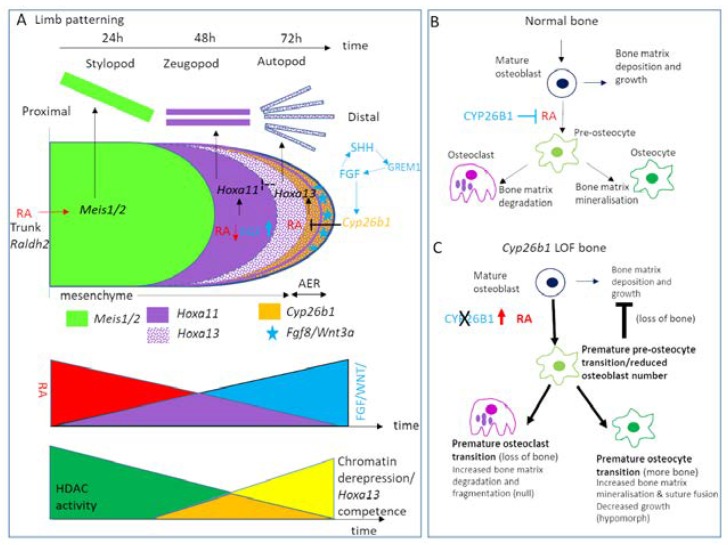
Roles for CYP26 enzymes in limb patterning and chondrogenic development. (**A**). Expression of *Cyp26b1* in the apical ectodermal ridge and underlying mesenchyme induced by FGF8 signalling is required to degrade RA synthesised by RALDH2 in the trunk of the embryo near the outgrowing limb bud. This establishes an opposing proximodistal (PD) RA gradient to the FGF/WNT PD gradient. High RA is required to specify the most proximal limb element the stylopod by inducing *Meis1/2* expression. Reduced RA levels at a point along the PD gradient inhibit *Meis* expression and in conjunction with FGF signals, promote expression of distal *Hox* gene *Hoxa11*, necessary to specify the zeugopod. Further reduction of RA levels distally by CYP26B1 degradation permits more distal *Hoxa13* expression, once time-dependent HDAC activity is sufficiently reduced to allow chromatin derepression. This mediates competence for *Hoxa13* expression in the limb mesenchyme and thus specification of the most distal limb element, the autopod. Adapted from Rosello-Diez et al. 2014 [204] (**B**). Role of *Cyp26b1* in regulating RA signalling to prevent premature transition from mature osteoblasts producing bone matrix to bone mineralising osteocysts and bone degrading osteoclasts. (**C**). Loss of RA degradation by CYP26B1 leads to a range of phenotypes resulting from premature osteocyte and osteoclast differentiation including loss of bone due to decreased osteoblasts producing bone matrix and increased numbers of bone degrading osteoclasts plus increased osteocytic bone mineralisation. Adapted from Laue et al. 2011 [202].

**Figure 7 jdb-08-00006-f007:**
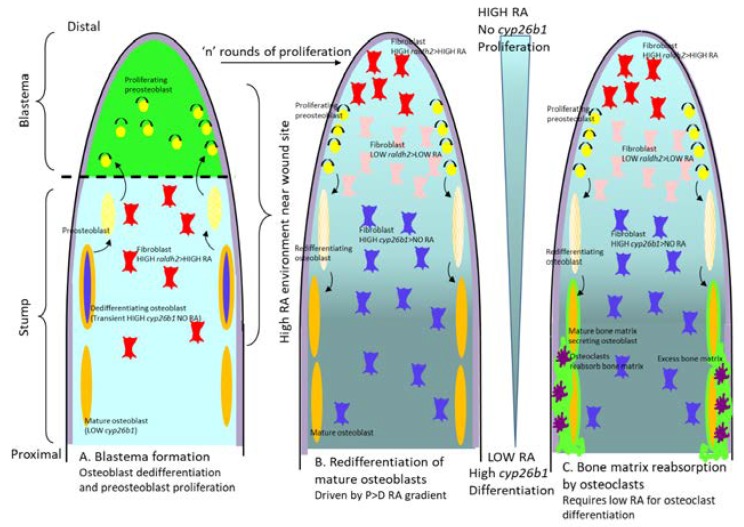
Cell autonomous and cell-non autonomous regulation of RA availability by CYP26 enzymes during zebrafish fin regeneration. (**A**). Blastema formation. A high RA environment produced by RALDH2 RA synthesis in stump fibroblasts near the amputation site is necessary to promote proliferation of preosteoblast cells forming the blastemal in response to the injury. However, dedifferentiation of mature osteoblasts is necessary to give rise to initial preosteoblasts and this requires a low RA environment. These two opposing requirements are met by a transient high level of *Cyp26b1* expression in mature osteoblasts (cell autonomous function). This produces a low level of RA in osteoblasts allowing dedifferentiation to preosteoblasts, whilst maintaining the high RA environment necessary for preosteoblasts to proliferate. (**B**). After several rounds of preosteoblast proliferation, these cells need to initiate redifferentiation towards mature bone matrix producing osteoblasts if the amputated fin is to regenerate. This respecification requires the establishment of a proximal to distal RA gradient. This is generated by altering the expression of the enzymes controlling RA availability in the fibroblasts of the regenerating stump. Distally, fibroblasts maintain *Raldh2* expression and thus synthesis of RA, allowing continued preosteoblast proliferation by maintaining a high RA environment. More proximally, fibroblasts express only low levels of Raldh2 leading to lower paracrine RA levels, thus allowing the redifferentiation of preosteoblasts towards mature osteoblasts to begin. Most proximally, fibroblasts express high levels of *Cyp26b1* thus degrading RA, promoting an RA-free environment suitable for the maintenance of mature bone-matrix producing osteoblasts in the regenerating stump. (**C**). This low/negative RA environment is also required for the differentiation of osteoclasts from osteoblasts. Osteoclasts degrade excess bone matrix deposition, and seem required in the amputated fin to help mediate correct bone regeneration. Adapted from Blum and Begemann et al. 2015 [371].

**Table 1 jdb-08-00006-t001:** Summary of Phenotype Causing Mutations in Human *CYP26* genes.

Gene	Nucleotide/AA Change	Change in Enzyme Function	Phenotype	Reference
*CYP26A1*	R173S	?	None reported	[206]
	F186L	40–80% reduced atRA metabolising activity in COS cells		
	C358R			
*CYP26A1*	g.3116delT	reduced atRA metabolising activity	Associated with spina bifida	[208]
	premature stop			
*CYP26A1*	rs4411227 C/G or C/C	?	Increased risk oral and pharyngeal cancer	[209,210,211]
*CYP26A1* and *CYP26C1*	Microdeletion of up to 249–363 kb of chrs. 10q23.33	Haploinsufficiency	Optic nerve aplasia	[213]
		*CYP26A1*, *CYP26C1*, *EXOC6*		
*CYP26A1* and *CYP26C1*	8.3 Mb microdel. Chrs 10q23.2–23.33. The 79 deleted genes included CYP26A1 and C1,	Haploinsufficiency *CYP26A1*, *CYP26C1* + 79 other genes	Premature ageing skeletal and dental development, retinal scarring, and autism-spectrum	[214]
		Raised plasma RA levels		
*CYP26B1*	Nine missense or splicing changes	100% loss of function	Neural tube, limb, craniofacial, skeletal, heart, kidney and lung defects	[207]
	(samples collected during gestation)			
	583C > T Arg195Met (1)			
	589c > A Leu197Met (3)			
	704G > A Arg235Gln (1)			
	712C > G Gln238Glu (1)			
	715G > A Ala237Thr (3)			
*CYP26B1*	Splicing variant with loss of exon 2	30% loss of function	Expressed in atherosclerotic lesion vascular cells	[215]
*CYP26B1*	rs3768647/9309462	?	Increased risk oral and pharyngeal cancer	[209,210,211,212]
	C/C or C/T			
	rs138478634 G/A change in exon 5			
*CYP26B1*	rs2241057T/T (major allele	Lower CYP26B1 activity	Increased risk of Crohn’s disease	[217]
	rs2241057C/C (minor allele)	Higher CYP26B1 activity	Larger macrophage-rich atherosclerotic	[216]
	L264S		lesions	
*CYP26B1*	3 c.1088G > T	100% loss of function by affecting the K-helix	Craniofacial, skull, pelvic, limb long bone skeletal defects	[202]
	homozygous			
	p.Arg363Leu			
	1 died in utero/2 terminations			
*CYP26B1*	c.436T > C	31% loss of function	Defects similar to Antley–Bixler and Pfeiffer Syndromes	[202]
	homozygous		Skull, digit and joint skeletal defects	
	p.Ser146Pro			
	Died at 5 months			
*CYP26B1*	c.1303G > A	Predicted loss of function	Skull and long bone skeletal defects, intellectual disability	[218]
	p.Gly435Ser			
	homozygous			
	(survived to adulthood)			
*CYP26B1*	0.78–4 Mb microdeletion 0.78–4 Mb chrs.2p13.2–13.3	8% induction of CYP26B1 mRNA by RA compared to controls	Intellectual disability, language delay, hyperactivity, dysmorphic facies and vertebral and/or craniofacial abnormalities	[219]
	*CYP26B1* and *EXOC6B* haploinsufficient			
*CYP26C1*	Missense p.Phe508Cys	Loss of function	Confers increased severity of SHOX p.Val161Ala mutation skeletal and short stature phenotype	[223]
*CYP26C1*	Missense c.148C > T, Pro50Ser;	Loss of function	Short stature phenotype	[224]
	c.356A > C, p.Gln119Pro; c.910G > A, Ala304 The			
	Splice variant truncation			
	c.706-A > C			
*CYP26C1*	Duplication > frameshift	100% loss of function	Focal facial dermal dysplasia	[225]
	> premature stop			
	c.844–851dupCCATGCA			
	p.Glu284*fs*X128 (maternal)			
	homozygous and compound heterozygous			
	Missense c.1433G > A p.Arg478His (paternal)			
	compound heterozygous			
*POR*	Missense, nonsense, frameshift, splicing and exon deletions	Loss of function	Antley–Bixler like skeletal defects and deficient steroidal profiles	[220,221,222]

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
