# Peer review of "Regulating Retinoic Acid Availability during Development and Regeneration: The Role of the CYP26 Enzymes"

_jdb, 2020, doi:10.3390/jdb8010006_

Round 1
Reviewer 1 Report
This review article is well written and broadly covers topics related to retinoic acid metabolism. There are a few comments and suggestions for the improvement.
1) Through this article, there are several typo so that it is better to proofread and edit.
2) Line 58 -61: Need more information for each mutant mice.
3) Line 136-137: An error in sentence space.
4) Line 389-397: No need to use bold.
5) Figure 3: Need high resolution images.
6) There is few sentence describing craniofacial phenotype in mutant mice while it has been well studied for the role of retinoic acid metabolism in craniofacial development.
Reviewer 2 Report
The manuscript by Roberts is a thorough and well-written review of RA signaling in development. The review nicely focuses on the CYP26 family of genes as major RA metabolizing enzymes, with extensive discussions of their structure-activity, expression patterns, and evolutionary conserved roles in development. I believe the manuscript warrants publication and have only a few minor comments below:
Lines 194-196 are confusing, perhaps due to wording. I suggest rewording or adding additional details to clarify. Lines 199-202 are confusing and difficult to follow. Lines 309-313 are again confusing. Can the ideas grouped in this sentence be separated into separate sentences and linked? Figures 3 and 4C, perhaps just due to compression of uploaded files, are pixelated and difficult to read. Figure 1 has elements that are, to me, aesthetically unappealing. Most notably the line representing the cell membrane and some of the arrows indicating reaction pathways. This is of course personal preference, but I would recommend improving the quality of its presentation.Author Response
please see attachment
